# Predicting antifolate resistance in the unculturable fungal pathogen *Pneumocystis jirovecii*

Francois D. Rouleau[1,2,3,4,5,6]*, Alexandre K. Dubé[1,2,3,4,5,6], Alicia Pageau[1,2,3,4,5,6], Lyne Désautels[7], Philippe J. Dufresne[7], Christian R. Landry[1,2,3,4,5,6]

**1** Institut de Biologie Intégrative et des Systèmes (IBIS), Université Laval, Québec, Canada, **2** Département de Biochimie, Microbiologie et de Bioinformatique, Faculté des Sciences et de Génie, Université Laval, Québec, Canada, **3** PROTEO - Regroupement Québécois de Recherche sur la Fonction, l'Ingénierie et les Applications des Protéines, Canada, **4** Centre de Recherche en Données Massives (CRDM_UL), Québec, Canada, **5** Infectious Disease Research Center (CRI), Université Laval, Québec, Canada, **6** Institut intelligence et données (IID), Université Laval, Québec, Canada, **7** Laboratoire de santé publique du Québec (LSPQ), Institut national de santé publique du Québec (INSPQ), Sainte-Anne-de-Bellevue, Canada

* Francois.rouleau.2@ulaval.ca

## Abstract

*Pneumocystis jirovecii* is an opportunistic fungal pathogen responsible for Pneumocystis pneumonia (PCP) in immunocompromised patients. Antifolate drugs targeting the dihydrofolate reductase (DHFR), including trimethoprim (TMP), remain central to treatment, but studying the effects of mutations in DHFR on resistance to treatment is limited by our inability to culture this organism *in vitro* or in animal models. We expressed *P. jirovecii* DHFR (PjDHFR) in *Saccharomyces cerevisiae* and performed deep mutational scanning (DMS) on this protein to measure the effects of all single amino-acid substitutions on enzyme function and resistance to methotrexate (MTX), a model antifolate which shares structural features with TMP. We integrated experimental results with structural and evolutionary features from multiple biophysical modeling approaches, and by using an interpretable machine-learning framework, we trained a random forest model to classify MTX resistance-conferring mutations in PjDHFR. We then leveraged this framework as a prediction tool to model the effects of mutations on resistance to TMP, which cannot be directly assayed experimentally. Functional measurements from DMS were the strongest contributors to resistance prediction and generally outperformed purely computational features. Resistance-conferring mutations were constrained by function, revealing a functional–resistance trade-off within this essential protein. Feature contribution analyses highlighted key predictors such as distance to ligand, flexibility, stability, and functional trade-off as determinants of resistance. When extrapolated to TMP, the model identified candidate resistance mutations consistent with known biochemical constraints of DHFR. We demonstrate how experimentally measured functional landscapes can be combined with biophysical modeling to help understand and predict antifolate resistance in an unculturable

**Data availability statement:** The sequencing data produced for this work is accessible through BioProject accession PRJNA1243435. Raw read sequencing from PjDHFR sequencing from patient samples is available at BioProject accession SAMN47937957. All scripts for figures and analysis, as well as processed data, are available on GitHub at https://github.com/Landrylab/Rouleau_et_al_2025.git. Detailed files and structures for FlexddG analysis are available upon request, as the complete output is around 4.8Tb of data (~4.2M structures + accompanying files). FlexddG input and output files are available at Zenodo DOI: 10.5281/zenodo.15269626.

**Funding:** The project was funded by a Genome Canada and Genome Quebec grant (6569, https://genomecanada.ca/). CRL holds the Canada Research Chair in Cellular Systems and Synthetic Biology (https://www.chairs-chaires.gc.ca/home-accueil-eng.aspx). FDR was supported by fellowships from FRQNT, PROTEO, EvoFunPath NSERC CREATE program and the Vanier Canada Graduate Scholarship agency. AP and AKD received a salary covered by the Genome Canada and Genome Quebec grant (6569, https://genomecanada.ca/). LD and PJD received a salary covered by INSPQ and the Gouvernement du Québec. The funders had no role in study design, data collection and analysis, decision to publish, or preparation of the manuscript.

**Competing interests:** The authors have no competing interests to declare.

fungal pathogen. Our results provide biological insight into the constraints affecting the evolution of resistance in PjDHFR, and support that resistance arises from mutations altering drug interactions while preserving function. We illustrate how DMS data can enable generalizable, mechanistically interpretable models of drug resistance across structurally related antifolates.

## Author summary

*Pneumocystis jirovecii* is a fungal pathogen causing pneumonia in immunocompromised humans. Infections by *P. jirovecii* are treated using drugs that prevent this pathogen from making folate, an essential component of many cellular mechanisms. In recent years, this treatment has been failing in an increasing number of cases, implying the evolution of resistance to this treatment. As *P. jirovecii* does not grow in the lab, the investigation of this resistance has been difficult, and common lab models do not respond to the drugs used to treat it. To overcome these limitations, we use a combination of experimental data and computer modeling to train a machine learning model to predict how genetic changes in one of the drug targets might cause drug resistance in this pathogen. The presented model predicts mutations in the drug target that may make this pathogen resistant to treatment, including mutations that have been previously characterized *in vitro* as drastically reducing drug binding. To investigate if our model predicted mutations occured in nature, we also sequenced the largest number of this pathogen's drug target to date. Our study provides new tools to predict drug resistance in hard-to-study pathogens, helping to understand and potentially respond to treatment failure.

## Introduction

The number of human pathogens is estimated to be around 1400 species, with approximately 300 being fungi [1,2]. One such pathogen is *Pneumocystis jirovecii*, a human-specific obligate parasitic commensal, and opportunistic pathogen that causes Pneumocystis pneumonia (PCP) in vulnerable patients [3]. In healthy patients, *P. jirovecii* usually does not cause PCP, and rather leads to asymptomatic infection and transient colonization. Environmental exposure to *P. jirovecii* is frequent, with reports from some countries indicating it can be present in up to 40% of individuals in a population [4–6]. This pathogenic fungus has also been known to cause PCP in HIV-negative infants [7]. Historically, PCP has been most prevalent in HIV-positive individuals, but with the advent of efficient antiretroviral therapies, its occurrence has been decreasing [8]. However, infection rates have been increasing in immunocompromised, non-HIV populations [9,10]. *P. jirovecii* has recently been placed on the WHO fungal pathogen priority list [11].

Research on *P. jirovecii* has been historically difficult, as this pathogen has not been reproducibly culturable *in vitro* [12]. *Pneumocystis* species are not known to have an ecological niche outside of mammal lung epithelial tissues, and all known species are host-specific, with no known reports of transmission of a given *Pneumocystis sp.* outside of its associated niche [13,14]. *P. jirovecii* has the peculiarity of being insensitive to most classical antifungals, as they have not been found to efficiently treat PCP [15]. As such, the antifolate combination drug Trimethoprim-Sulfamethoxazole (TMP-SMX, Bactrim), is used as the mainstay drug for both prophylaxis and treatment against PCP. These drugs target the dihydrofolate reductase (DHFR) and the dihydropteroate synthase (DHPS), respectively [16,17]. In the last two decades, instances of TMP-SMX treatment failure have been increasing in PCP cases, implying the evolution of resistance in this fungus [18–23]. There is therefore a need to investigate mechanisms of resistance, and to gain insight into the mode of action of this drug combination in fungal infections. Understanding the molecular basis of antifolate resistance and the evolutionary constraints that shape it, is therefore critical for both clinical management and the anticipation of future resistance trajectories.

Historically, treatment failure has been associated with mutations in the DHPS enzyme, the target of SMX. However, an increasing number of studies have also reported mutations in DHFR when failure of TMP-SMX treatment is observed [24–27]. Furthermore, reports from eastern Canada, where 265 DHPS from clinical samples have been sequenced, only found DHPS mutations associated with treatment failure in three samples (~1.5%), with those mutations very rarely observed in patients with TMP-SMX treatment failure as well, suggesting a greater potential contribution from the DHFR to resistance than previously estimated, or the presence of a different mechanism of resistance (data available at Laboratoire de santé publique du Québec (LSPQ)). These observations raise the question of how resistance-conferring mutations in DHFR arise, and whether their effects may generalize across antifolate drugs targeting the same enzyme, as well as the constraints affecting the evolution of such mutations.

As there are currently no susceptibility assays available for *Pneumocystis jirovecii*, our ability to study the evolution of resistance in this pathogen is limited to biochemical assays *in vitro* and *in silico* approaches [12]. An early study focused on testing *in vitro* biochemical constants, such as the inhibitory constant ($K_i$), the Michaelis constant ($K_m$) and the catalytic constant ($k_{cat}$) of protein variants that had been identified in clinical studies in patient samples where TMP-SMX treatment had failed. Their experiments identified that these variants had decreased affinity for TMP. However, the effect of mutations on the protein canonical function *in vivo* could not be validated, limiting the broader conclusions of this study [24]. Another study focused on using computational tools to identify the effects of mutations on resistance. Using molecular dynamics to model the effects of mutants characterized in [24], the authors used a machine learning model to classify the mutations in sensitive/resistant classes. The resulting model accurately classified variants in ~65% of cases and relied on the two most predictive features identified through principal component analysis. However, their conclusions were limited by the small amount of *in vivo* information available, as well as computational constraints, since extensive molecular dynamics were necessary for each variant tested. This only allowed them to focus on 19 PjDHFR mutants, and again without knowledge on the effect of mutations on protein function, limiting the broader conclusions of this study [28]. Together, these studies highlight the difficulty of dissecting resistance mechanisms from protein function in the absence of experimental functional data, as well as the limitations of predictive models based purely on computational modeling.

Given the current limitations, there is a need to find ways to test mutational effects, ideally in a context where the impact of mutations on DHFR function can also be assayed. We recently used functional complementation of DHFR in *S. cerevisiae* to systematically investigate the effect of mutations on resistance to antifolates through mutations in *P. jirovecii*'s DHFR (PjDHFR) [29]. Because *S. cerevisiae* is not sensitive to TMP, we could not investigate the effects of mutations in PjDHFR on TMP resistance *in vivo*. However, we used deep mutational scanning (DMS) of PjDHFR to identify mutations that caused resistance to another similarly competitively inhibiting antifolate molecule, methotrexate (MTX). We reasoned that if we could use computational tools to predict MTX resistance accurately using this experimental data as ground truth, we could use the same tools to predict resistance to TMP. Indeed, as both MTX and TMP are antifolate

competitive inhibitors of the same protein, and share molecular properties and binding site, we expect that resistance mechanisms should share characteristics as well [30–32]. However, the extent to which resistance determinants transfer across antifolate drugs remains an open question and may be constrained by differences in molecular interactions and functional trade-offs.

Here, we leverage our previous DMS data on PjDHFR's resistance to MTX to train a machine learning model to predict resistance from the protein structure and properties. We use a suite of molecular features in the model, including an *in silico* analysis of the biophysical impacts of mutations on proteins [33–35]. One important piece of data for making these predictions are the effects of mutations on the canonical function of PjDHFR as mutations that appear to be susceptible to MTX or that could be predicted to decrease interaction affinity between the protein and MTX could in fact be inactivating or destabilizing the enzyme. We therefore performed an additional set of experiments to estimate the impact of all amino acid substitutions in PjDHFR on its ability to complement the *S. cerevisiae* DHFR. In addition to developing a predictive framework, we aim to quantitatively evaluate the extent to which experimental functional measurements improve resistance prediction relative to purely computational features. By explicitly comparing models trained with and without DMS functional data, we aim to validate if computational predictors alone are limited in their ability to capture the functional constraints of the evolution of antifolate resistance. Then, using the model trained to predict resistance to MTX and our experimental data as a reference data set, we use the model to predict resistance to TMP. By integrating functional measurements with structural and evolutionary features, our approach aims to predict the effects of mutations on resistance, but also to identify biological constraints and non-linear interactions that potentially govern the evolution and generalizability of antifolate resistance in an unculturable pathogen. Our computational and experimental approaches are outlined in Fig 1 [33–35].

## Results

### DMS of PjDHFR reveals functional constraints and conserved strongly deleterious effects hotspots

To measure the effect of mutations on protein function (Fig 1A), we screened every possible single amino acid mutant in PjDHFR in a temperature sensitive (TS) DHFR constructs (TS-DHFR), in four independent replicates (S1 Fig). We transformed the mutant library built on plasmids in each replicate and grew each for approximately 6 generations. The screen was conducted in two conditions: 20°C was set as the permissive condition, and 37°C as the selection condition for the TS-DHFR strains, where the genomic DHFR should be fully inactivated. The 20°C condition was selected to minimize the effect of heat shock and incubation on TS-DHFR strain growth, and we expected small effects from the PjDHFR library in this condition.

Mutation effects were first measured on a per-codon basis, and correlation between replicates and fragment overlaps was evaluated by codon, then normalized by the fitness of the wild-type sequence (S2A, S2B Fig, Eq. 1). We found that selection coefficients were correlated between fragment overlaps in both temperature conditions (Pearson's $0.91 < r < 0.98$, mean of 0.94). Once we observed that the replicates were correlated in the selection condition (S2 Fig, Pearson's $0.84 < r < 0.94$, mean of 0.88), codons were grouped by amino acid at each position across replicates, with the median score per amino acid substitution used as selection coefficient for subsequent analyses (Fig 2A). The resulting score, now standardized between fragments and replicates, was used as the selection coefficient for subsequent analyses. A heatmap of the selection coefficient for the screen at the permissive temperature (20°C) for TS-DHFR strains is available in S3 Fig. At 37°C, we observed that most mutations had either a neutral or deleterious effect on protein function (selection coefficient of $\leq 0.0$), and that most positions where several mutations led to deleterious effects were in contact with one or both DHFR's canonical substrates, folate (DHF) and nicotinamide adenine dinucleotide phosphate (NADPH) (Fig 2A).

We observed a correlation between the selection coefficient from the MTX resistance screen, and selection coefficients for function measured at 20°C (Spearman's $\rho = 0.44$, $p < 1e{-}100$, Fig 2B) and 37°C (Spearman's $\rho = 0.52$, $p < 1e{-}100$, Fig 2C). Mutants behaved similarly on both temperature conditions (Spearman's $\rho = 0.78$, $p < 1e{-}100$, Fig 2D), implying that

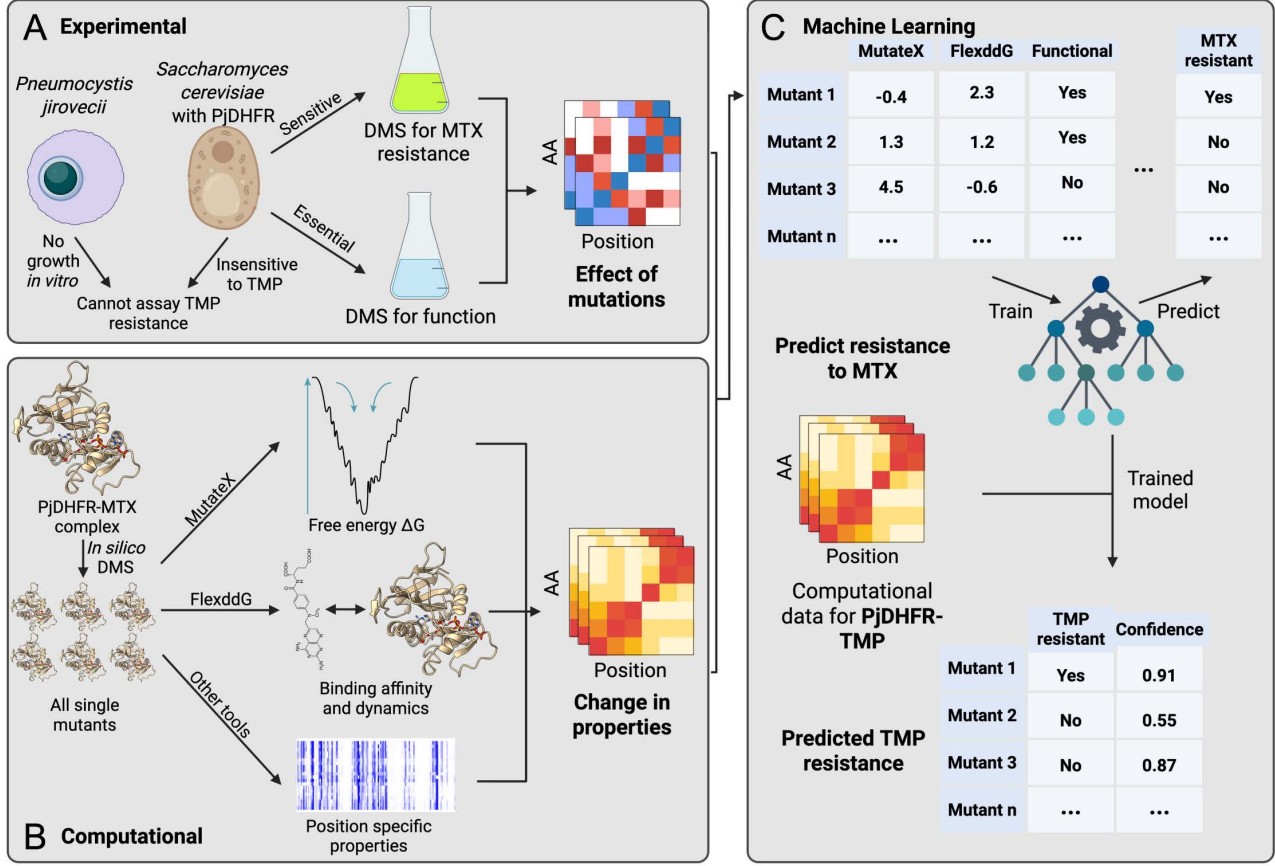

**Fig 1. General workflow of the approach to predict drug resistance to TMP in *Pneumocystis jirovecii*. A)** Using *S. cerevisiae* strains where the DHFR deletion was functionally complemented with the *P. jirovecii* ortholog (PjDHFR), we measured the impact of all amino acid substitutions on both function and MTX resistance. This is used downstream as ground truth when training the model. **B)** Using PjDHFR-MTX 3D structure from [36] (Alpha-Fold model UniProt A0EPZ9), we used modeling suites to conduct *in silico* DMS to estimate the effect of mutations in different ligand-dependent and independent effects. These data were used to train a random forest model to predict MTX resistance based on the different mutation properties. **C)** The model was used on similar data generated using the PjDHFR-TMP complex, and predictions were made on the effect of mutations in PjDHFR resistance to TMP, which cannot be tested experimentally. Panel A, B and C were created in BioRender. 1, **U.** (2025) https://BioRender.com/2ja2o1o.

even the low temperature transformation protocol and incubation at 20°C caused fitness defects in the TS-DHFR strains. However, as all biological, codon and fragment overlap replicates are well correlated, we expect that this effect had minimal consequences on selection coefficient in selection conditions (S2 Fig). After grouping the data from both assays, we covered 3758/3914 (96.0%) of all possible amino acid substitutions with selection coefficients for both function and MTX resistance.

Most mutations had little to no detectable effect on protein function. Indeed, 3140/3758 amino acid substitutions (83.5% of covered mutations) were grouped as being wild-type-like in the Gaussian mixture model (selection coefficient ≥ -0.124 at 37°C, S4A Fig), and therefore having a neutral effect on protein function at this expression level and in these experimental conditions. All tested positions had at least one substitution with a neutral effect, except position G125, where all substitutions were deleterious. Across 204 amino acid positions, there were 71 positions where all substitutions had a selection coefficient ≥ -0.124, indicating that all mutations at these positions had a neutral effect and that these positions are highly tolerant to amino acid substitutions. Overall, the median number of neutral substitutions per position was 18/19, indicating that most positions are tolerant to amino acid substitutions.

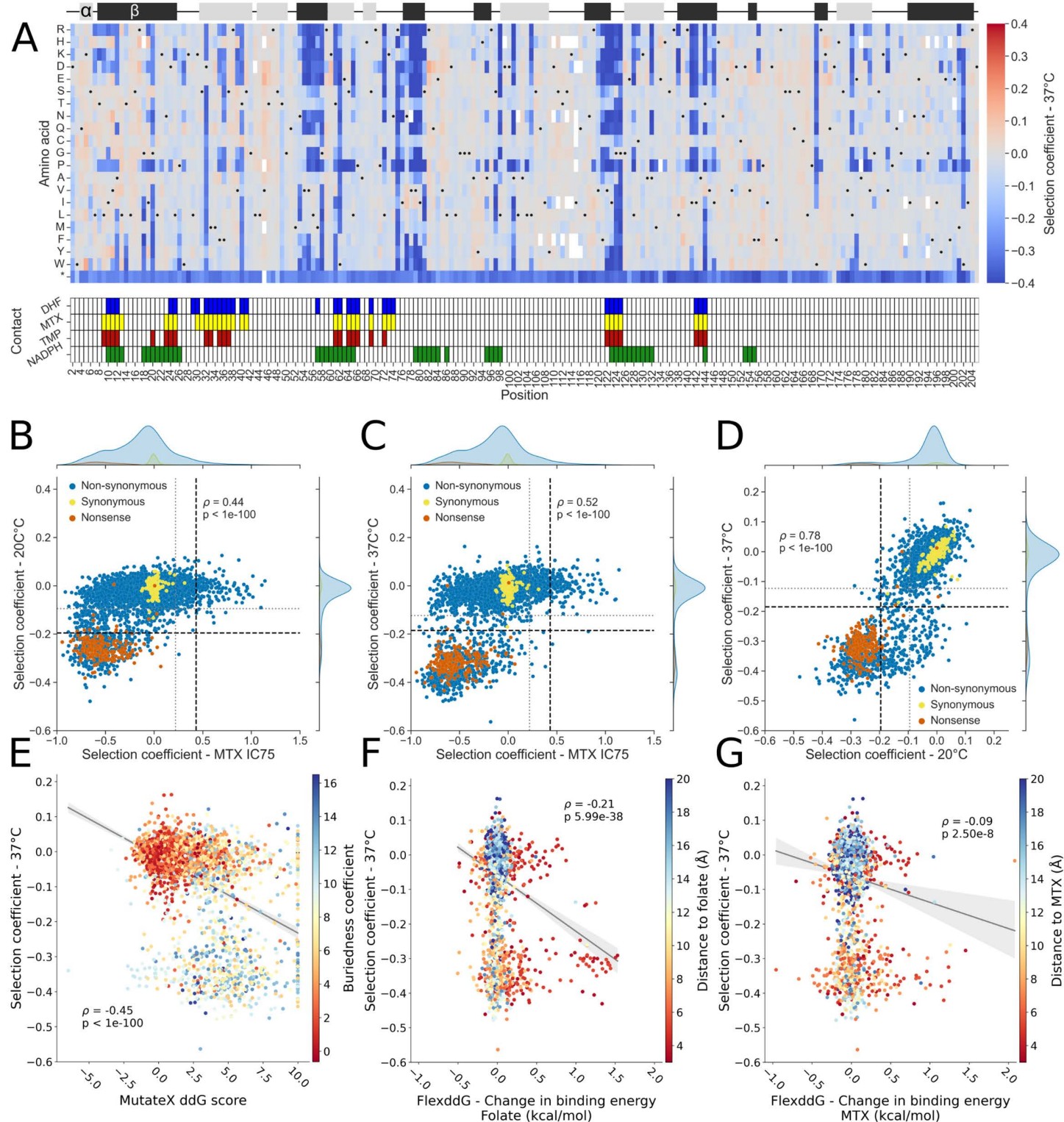

**Fig 2. Effects of amino acid substitutions on PjDHFR function as assayed in *S. cerevisiae* TS-DHFR strains. A)** Heatmap of selection coefficients at 37°C, where TS-DHFR alleles are non-functional. Selection coefficient is measured in relation to the median of silent mutations. Black dots within the heatmap represent the wild-type sequence. The track at the top of the heatmap represents secondary structures, with α-helices in light grey and

beta strands in dark grey. The track below the heatmap represents contact between the different ligands and the residues in the protein. Contact was defined as amino acids with an α-carbon located less than 8 Å from MTX, DHF, TMP or NADPH when the ligand is within its binding pocket (PDB: 3CD2 (MTX and NADPH) and 4CD2 (DHF)). **B)** Comparison MTX screening at IC75 from [36] and screening done at 20°C (Spearman's $\rho = 0.44$, p < 1e-100). Horizontal lines are thresholds for function established from the mixture models (grey line, partially functional ≤ -0.09, non-functional, black line, ≤ -0.195) and vertical lines are thresholds from [36]. **C)** Comparison MTX screening at IC75 and screening done at 37°C (Spearman's $\rho = 0.52$, p < 1e-100, grey line, partially functional ≤ -0.124, non-functional, black line, ≤ -0.185). Graphs B) and C) were made to ensure that no mutations were identified as being both resistant and non-functional. Such mutations would fall in the bottom right quadrant. **D)** Comparison between screening done at 20°C and 37°C (Spearman's $\rho = 0.78$, p < 1e-100). Grey and black lines are as in panels B and C. **E)** Comparison between computed MutateX changes in free energy for all mutants and screening done at 37°C, colored by residue buriedness, (Spearman's $\rho = -0.45$, p < 1e-100). **F)** Comparison between computed FlexddG changes in interaction energy between binding energy to DHF and screening done at 37°C, colored by proximity of α-carbon to the closest heavy atom on the ligand for residues in contact with the ligand (<8 Å, Spearman's $\rho = -0.21$, p = 5.99e-38). **G)** Comparison between computed FlexddG changes in interaction energy between binding energy to MTX and screening done at 37°C, colored by proximity of α-carbon to the closest heavy atom on the ligand <8 Å, Spearman's $\rho = -0.09$, p = 2.50e-8).

We observed 583 amino acid substitutions as having deleterious effects with a high degree of confidence (14.2% of substitutions, selection coefficient < -0.185, p-value < 0.05, Benjamini-Hochberg (BH) corrected, S4 Fig), causing large growth defects when grown at 37°C in the TS-DHFR backgrounds. We define these amino acid substitutions as having strongly deleterious effects in this assay (SDE) mutations (Fig 2A-2C, mutants below dotted black lines). Most positions with such strong deleterious effects often had many mutations leading to SDE, with very few positions where only one mutation caused strongly deleterious effects, except in the case of substitutions towards proline. Interestingly, 39.9% SDE mutations are found within a radius of 9 Å of each other (S5 Fig). This structured region, composed of a β-sheet and an α helix from positions 54–64 (β-strand into α-helix), 75–81 (β-strand) and 121–126 (β-strand), is in close proximity to both the folate and NADPH binding pockets.

To support these experimental observations, we computed GEMME scores, which is a score used to predict mutational effects based on sequence conservation across orthologs, where a lower score represents a substitution that is more likely to be deleterious [37]. This region has statistically significantly more negative GEMME scores than the rest of the protein (Mann-Whitney U rank test, p < 0.05), validating its high degree of importance for protein function (S5 Fig). In PjDHFR in selective conditions, only 58 mutations (1.4% of amino acid mutants, -0.124 > selection coefficient > -0.185, p-value < 0.05, BH corrected) were identified as having statistically significant intermediate effects, with all other mutations being non-significantly different from either wild-type residues or stop codons. Interestingly, for most positions with low tolerance to mutations, the conservation of amino acid properties appeared to be key to retain protein function, as opposed to the presence of a specific amino acid, as shown in previous studies [38,39]. The 14.2% of mutations causing SDE in PjDHFR (206 aa) is comparable to *folA*, the DHFR ortholog in *Escherichia coli* (160 aa), where 12% of missense mutations lead to SDE [40], consistent with the high degree of structural conservation in orthologous DHFRs [41]. These regions are also known to be critical for function in *folA* [42]. Evolutionarily conserved positions with low tolerance to mutations, such as positions G20, R75 and G125 in PjDHFR (G15, R57 and G96 in *folA*) are also sensitive to most amino acid substitutions in *folA* [40]. While these observations are consistent with established principles linking sequence conservation and sensitivity to mutations, they provide a necessary functional baseline for interpreting resistance phenotypes within proteins. This systematic genotype-phenotype mapping had never before been conducted at this throughput within a eukaryotic DHFR.

For a mutation to lead to resistance, some level of protein function must be maintained. Indeed, even if a mutation would prevent binding between PjDHFR and MTX, if that mutation also prevents protein function, it will not lead to resistance, as it would not allow for cell survival or proliferation. Our dataset makes this functional–resistance constraint explicit, as most mutations that affect function led to SDE (Spearman's $\rho = 0.52$ between function and resistance at 37°C, p < 1e-100). However, as our thresholds for calling functional/non-functional (Function selection coefficient ≥ 0.185) and resistant/sensitive (MTX resistance selection coefficient ≥ 0.550 at IC90) mutants in our experiments were set using

Gaussian mixture models (S4A, S4B Fig), some mutants close to either threshold could be classified as non-functional and resistant, which would imply a tradeoff. These include mutants I10D, G124C, G124D, F199D and F199G. Across the entire protein, only 11 mutations (0.2% of amino acid mutants, DMS score -0.124 ≥ at 37°C and ≥ 0.220 in MTX condition, p-value < 0.05) resulted in a partially functional protein that showed resistance to MTX (Fig 2B-2C, bottom right of graphs), with only two mutations (I10D and F199D) being identified as conferring high levels of resistance when setting thresholds using Gaussian mixture models. The rarity of such tradeoff mutations indicates that resistance evolution in PjDHFR is strongly constrained by the requirement to preserve enzymatic activity, consistent with the essential nature of this protein and the mechanisms of inhibition of antifolates. Otherwise, mutations that lead to MTX resistance appear to have little to no effect on canonical protein function in the assay based on functional complementation of *S. cerevisiae*'s DHFR. This is similar to recent results obtained when studying *ERG11*, another generally essential gene targeted by antifungal drugs, where limited tradeoff was observed between function and resistance [43].

## Complementary modeling approaches help explain functional effects of amino acid substitutions

No single modeling method can yet recapitulate all aspects of protein dynamics and function. We can therefore expect that different modeling methods will each explain part of the experimental effects we observed during DMS screening. We mainly focused on two modeling tools which allow us to predict effects caused by mutations. The first tool, MutateX (v0.8) [32], is based on FoldX (suite 5) [43], and allows to measure the effect of a mutation on a protein's folding free energy, where mutations with positive ΔΔG are being predicted as being destabilizing, and mutations with a negative score as stabilizing. By using FoldX empirical force-fields to calculate the energy of both the wild-type and mutant structures for each amino acid mutant across the protein, as well as FoldX repair function to minimize clashes within the structure before computation, this software estimates the ΔΔG caused by mutations based on five independent simulations.

The second tool, FlexddG [33], allows to model the effect of mutations on the interaction free energy between two molecules and computes the changes caused by amino acid substitutions. It was used to compute changes in binding free energy between PjDHFR and its different substrates and inhibitors (Fig 1B.) FlexddG uses a Rosetta ensemble-based method to estimate changes of binding free energy (ΔΔG between protein and ligand) by sampling structural variations caused by mutations through a "backrub" method, minimization steps and side-chain repacking [34]. For each mutation, 35 000 backrub steps are conducted on 100 individual instances to generate a statistically robust distribution of ΔΔG scores, which are then averaged to yield the final estimate of interaction free energy changes.

We observed that what correlated best with protein function was the changes in folding free energy from MutateX (Fig 2E, Spearman's $\rho$ = -0.45, p-value < 1e-100). Highly buried residues often had higher change in folding free energy than less buried residues and are predicted to destabilize the protein to a greater extent, as expected. Using FlexddG, we also observed a correlation between changes in binding affinity between PjDHFR and DHF, and protein function, highlighting this method's ability to capture the effects of mutations on free energy of binding. When considering only residues in contact with the ligand, we found a weak but statistically significant negative correlation where mutations with a greater destabilizing effect are more deleterious (Fig 2F, Spearman's $\rho$ = -0.199, p-value = 5.55e-07). This correlation was weaker when comparing changes in binding energy between PjDHFR and MTX with function across the protein, implying that this method can accurately capture small differences between ligand molecules (Fig 2G, Spearman's $\rho$ = -0.09, p-value = 2.50e-8), such as a single heavy atom between DHF and MTX.

While correlation analyses provide an initial measure of association, they do not directly quantify the proportion of variance in functional scores explained by each modeling approach. To further investigate feature contribution, we performed linear regression analyses using each computational feature individually to predict experimentally measured functional scores and calculated the corresponding $R^2$. These analyses reveal that predicted changes in folding free energy from MutateX ($R^2$ = 0.206) explain the largest variance in functional scores among features, with residue buriedness ($R^2$ = 0.200) and change in region flexibility ($R^2$ = 0.102) being second and third best (S6 Fig). This comparison confirms that stability

is the strongest single predictor of function, but also demonstrates that no individual feature captures the full spectrum of mutational effects, as expected.

## Random forest model highlights importance of ligand interactions and protein functionality in resistance prediction

From various experiments and computational analyses, we believed that we had gathered comprehensive data on different aspects of protein function, resistance, and mechanisms. While individual features, such as predicted changes in protein stability, were identified as strong predictors of experimental functional scores, no single metric fully captured the non-linear interactions often governing resistance. We therefore sought to use a machine learning framework not solely to maximize predictive accuracy, but to integrate heterogeneous features and model the interactions among stability, ligand contacts, evolutionary conservation, and functional measurements. If this data could be used to predict resistance to MTX using machine learning (ML), we hypothesized that the resulting model could be used to predict resistance to TMP, as they share biophysical properties and mechanisms of action (Fig 1C). Based on the Gaussian mixture model from [36], mutations were classified as being either resistant (selection coefficient ≥ 0.550), or sensitive to MTX (selection coefficient < 0.550), and this was used as the ground truth for classification to train the ML model.

To predict resistance to MTX, we used experimental and computational features. The computational features can be split into two categories, ligand-dependent and ligand-independent. Details on how features were computed and gathered are available in the materials and methods section, and details about each feature is available in S1 Table. Ligand-dependent features are affected by the presence of a ligand bound to the protein (DHF, MTX or TMP), and are not intrinsic properties of the protein. They include the change in binding free energy to the ligand caused by the amino acid change, the effect of mutation on region flexibility, and distance between α-carbon of a residue and the closest heavy atom on the bound ligand. Ligand-independent features are intrinsic properties of the protein or rely solely on protein sequence and include change in folding free energy (computed in the absence of ligands), residue side chain length in the wild-type protein, residue buriedness (compound measurement of solvent accessibility and distance to protein outer residues), and evolutionary constraints. Experimental features include the effect of the amino acid changes on function (selection coefficient measured in TS-DHFR strains in the absence of drugs), whether a position is involved in catalysis, and allostery, and are ligand-independent features [44]. The dataset, composed of 3758 amino acid substitutions for which data was available for each feature, was split into training and testing sets in a 80:20 ratio with a fixed seed, and z-score normalized to put all data on a similar scale. Using the training set, different models were tested and optimized to find the best performing one (see materials and methods for details on model selection). To assess model performance, we used several evaluation tools, such as confusion matrices, receiver operating characteristic, precision-recall display (Fig 3A). We also examined the balanced accuracy score, defined as the sum of recall for all classes which allows for better estimate of model performance for unbalanced classes.

Overall, the best performing model was a Random Forest (RF), with balanced accuracy scores of 98.3% (2907/3005 substitutions accurately classified) on the training set and 88.3% (716/752) on the testing set (Fig 3A, see Model code on GitHub for detailed parameters). Mutants with probability of > 0.5 were predicted as being resistant (class 1), and mutants with probability ≤ 0.5 were predicted as sensitive (class 0). Among the 111 amino acid changes experimentally identified as leading to high levels of MTX resistance, only 4 were misclassified as sensitive by the model (false negatives, V11E, V11M, S37P and G124P). Using the SHapley Additive exPlanations explainer function, we identified the relative importance of features used for prediction in the model (Fig 3A) and investigated the classification decision for the misclassified mutations. Mutation V11E (resistance probability in MTX of 0.33, selection coefficient in MTX of 0.857, selection coefficient at 37°C of 0.065) and V11M (resistance probability in MTX of 0.36, selection coefficient in MTX of 0.745, selection coefficient at 37°C of 0.015) were predicted as being destabilizing, explaining most of their classification as non-resistant (S7A Fig). Mutation S37P (resistance probability in MTX of 0.36, selection coefficient in MTX of 0.760, selection coefficient

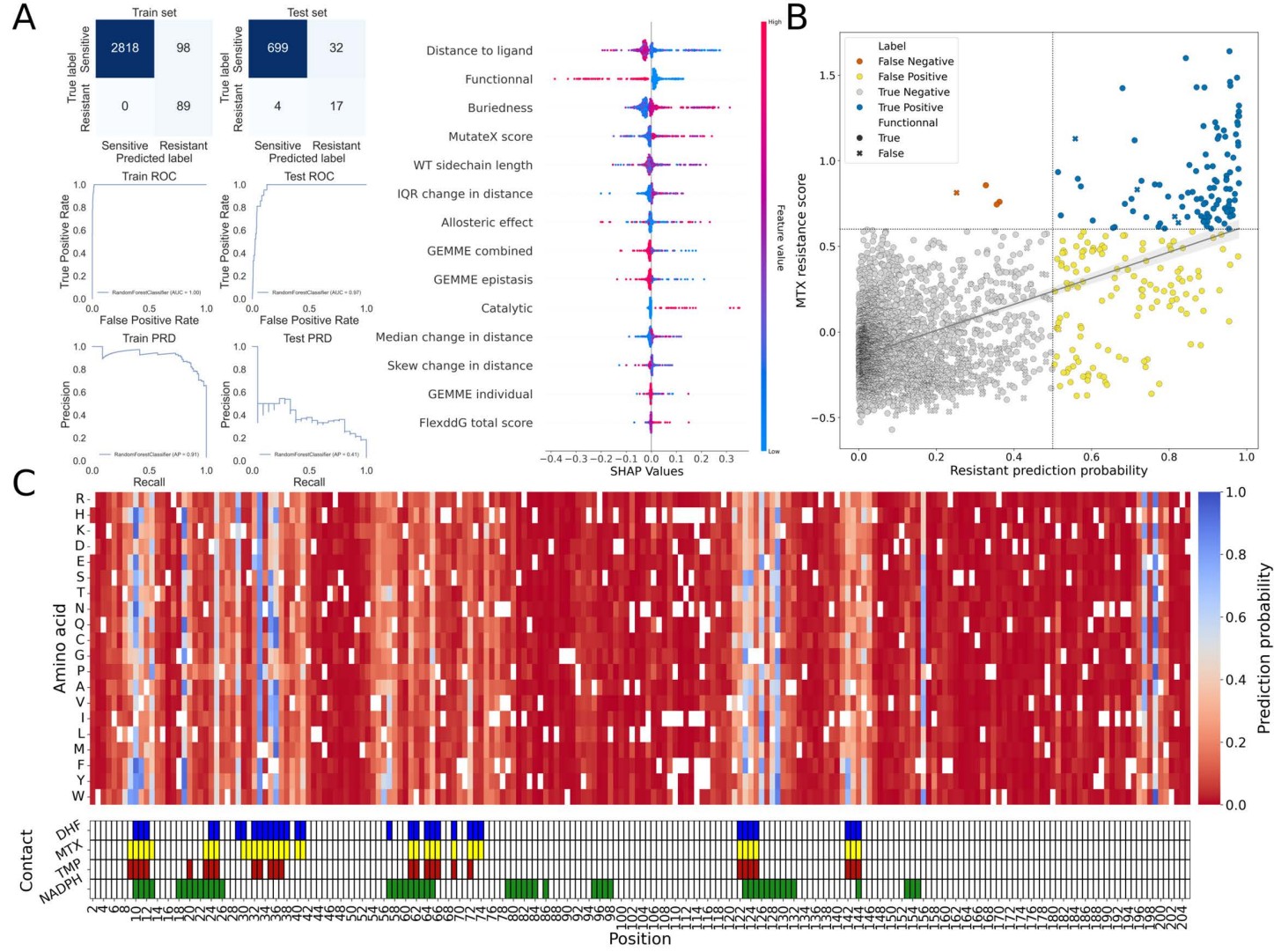

**Fig 3. Visualization of model performance. A)** Confusion matrices and ROC/PRD curves for train and testing set splits, with SHAP value explainer plot, showing relative impact of each feature on model decision. Features are ranked from top to bottom by feature importance in model decision. **B)** Scatterplot comparing ground truth from MTX DMS screening (MTX resistance score) vs model's predicted probability of being resistant (probability ranging from 0 to 1). True negatives are in grey, true positives are in blue, false positives are in yellow and false negatives are in orange. Mutations that were identified as functional in the function DMS screening have round markers (score > -0.185) and mutations identified as non-functional have X-shaped markers (score < -0.185). **C)** Heatmap with model's predicted probability of being resistant to MTX. Mutations with probability >0.5 are identified as resistance-causing, and mutations ≤ 0.5 are identified as sensitive. The track below the heatmap represents contact between the different ligands and the residues in the protein. Contact was defined as amino acids with an α-carbon located less than 8 Å from MTX, DHF, TMP or NADPH when the ligand is within its binding pocket (PDB: 3CD2 (MTX and NADPH) and 4CD2 (DHF)).

at 37°C of 0.039) was mischaracterized mainly because of its allosteric effect score, as it was scored as having an allosteric effect with high confidence in [45]. According to the model explainer, this allosteric score can have a strong effect on prediction in both directions (class 0 and class 1), depending on other position-specific factors. As for mutation G124P (resistance probability in MTX of 0.25, selection coefficient in MTX of 0.813, selection coefficient at 37°C of -0.325), it was identified as being non-functional, which is the major factor explaining its misclassification by the model (S7A Fig). Despite good overall performance, some effects could not be captured by the model.

PLOS Genetics

The feature which contributed most to successful overall classification of mutants was distance to ligand, which is coherent with results from [36], in which all but two positions where multiple substitutions lead to resistance were in contact with the antifolate inhibitor (Figs 2A and 3A).

Mutant functionality was the second most important feature in resistance prediction. Across all mutations identified as leading to MTX resistance, only 5 out of 111 were identified as being non-functional, with all 5 being close to the identified threshold between Gaussian distributions (Fig 3B-3C). Therefore, except in few cases near the margins, non-functionality is mutually exclusive to resistance. The misclassification of G124P supports that the model can capture this effect in most cases, classifying non-functional mutants as sensitive. When removing the experimental measures of SDEs as a model feature, we observed reduced model performance, showing an increase of ~50% in false positives across both sets (Balanced accuracy score of 96.9% on training set and 83.6% on testing set, S8A Fig). While this model was able to accurately classify most mutations that did not lead to MTX resistance (true negatives), it performed strictly worse overall than the model that includes experimental measurements of function. Assessing the effect of mutations on function is therefore a key feature to accurately predict the effect of mutations on resistance to antifolates.

Another important contributor of model decision was residue buriedness, where most residues with several resistance mutations fell in an intermediate range of buriedness, being neither deeply buried nor very exposed to solvent. Change in folding free energy, where mutations with a higher absolute change in folding free energy were often predicted as being sensitive was also a highly ranked feature. This is coherent with the negative correlation observed with function (Fig 2E). Interquartile range (IQR) of the change in distance between residue α-carbon and ligand heavy atom caused by mutations, a metric computed from FlexddG output structures, was also amongst the top predictive features. IQR of the change in distance is a measure of how much a given mutation affects the surrounding region's flexibility, where mutations that reduced flexibility were less likely to be predicted as resistance-causing. Interestingly, IQR of the change in distance also had a statistically significant negative correlation with the effect of mutations on function, meaning that mutations which reduce flexibility when bound to inhibitors more often have a deleterious effect on function (Spearman's $\rho$ =-0.24, p-value < 1e-100, S9 Fig).

While binary classification of MTX resistance based on Gaussian mixture models was used for classification, the measurements of MTX resistance were on a continuous scale. We therefore also used regression-based models to predict resistance score for each mutation (S10 Fig) and could compare the decision made by the best-performing classification model, and the selection coefficient predicted by regression models. Using the experimentally determined threshold from the screen for MTX resistance, we can consider mutants with a score >0.550 as being predicted as resistant by the regression model.

When using the simple linear regression model, no mutant is predicted as having a score of >0.550, which implies that no mutation can lead to MTX resistance, according to this model ($R^2$ =0.185, RMSE =0.236, Spearman's $\rho$ =0.326, p-value =1.213e-93, S10A Fig). When using the support vector regression (SVR) method ($R^2$ =0.323, RMSE =0.215, Spearman's $\rho$ =0.618, p-value < 1e-100, S10B Fig), predicted scores are skewed towards more negative values comparatively to ground truth (35/111 predicted as having a score of >0.550), indicating poor model fitting despite improvement on the linear regression model. Based on the best performance of the random forest classification method, we expected the random forest regression to outperform other regression-based methods, which is what we observed ($R^2$ =0.541, RMSE =0.177, Spearman's $\rho$ =0.864, p-value < 1e-100, S10C Fig). When compared to the random forest classification model, the regression-based model predicted less false positives, but more false negatives, with most predictions being close to the established classification threshold. We selected a random forest classifier, as it showed the best performance of the interpretable models. It could model complex, non-linear feature interactions while still allowing quantification of feature importance and partial dependence relationships. In contrast to more opaque models such as deep neural networks, this approach enables direct biological interpretation of how different structural and functional features contribute to resistance predictions. In this context, the ML model serves not only as a predictive tool but as an integrative framework to

probe the relative contribution and interaction of mechanistic determinants underlying antifolate resistance. The complete dataset used to train and test the ML models on PjDHFR's resistance to MTX, as well as the classification scores for all mutants using the best performing model, is available as S1 File.

**Ligand-specific features enable TMP resistance prediction from MTX-trained model**

Given the adequate performance of our model for MTX resistance, we applied it to predict TMP resistance (Figs 1C and 4) but this time using computational features modeled using the PjDHFR-TMP structure for predictions. As expected from feature importance during model training, mutations that are predicted to lead to resistance with high confidence are often mutations in close proximity to TMP heavy atoms, as well as positions where several mutations lead to MTX resistance. The model predicted 213/3758 (5.67%) mutations as leading to TMP resistance, with 74/3758 (1.97%) having a prediction probability of >0.8. Of all mutations predicted as leading to TMP resistance, 89/213 (41.78%) were experimentally identified as leading to high levels of MTX resistance, and 186 (87.32%) were predicted by the original model as leading to MTX resistance. We expected to find an overlap between mutations predicted to lead to MTX and to TMP resistance, as both antifolates share structural and mechanistic properties. The model also accurately captured the effect of functionality on resistance. Only three mutants were predicted as being TMP resistant while being identified as non-functional. All three mutants, G124C (Resistance probability in TMP of 0.55, selection coefficient in MTX of 0.831, selection coefficient at 37°C

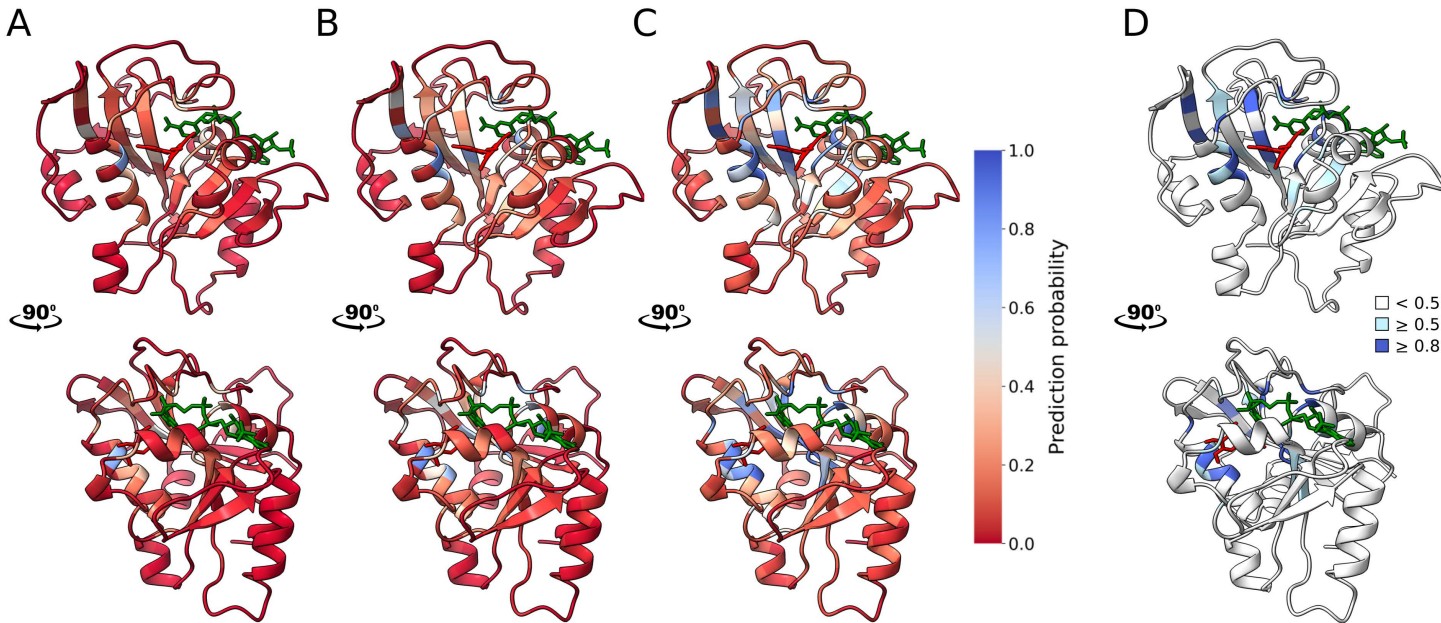

**Fig 4. Confidence score for mutations predicted to lead to TMP resistance mapped on PjDHFR's structure.** Using the model trained on resistance to MTX, we predicted the effect of mutations on TMP resistance using data from the different modeling strategies used on the PjDHFR-TMP complex structure. Mutations with probability ≥ 0.5 are identified as resistance-causing, and mutations < 0.5 are classified as sensitive. **A)** Position-wise minimum predicted probability of leading to TMP mapped on PjDHFR. TMP is colored in bright red, and NADPH in green. **B)** Position-wise median predicted probability of leading to TMP resistance mapped on PjDHFR. Ligand coloring is as in A). **C)** Position-wise maximum predicted probability of leading to TMP mapped on PjDHFR. Ligand coloring is as in A). **D)** Mapping of positions based on prediction probability of resistance thresholds. Positions with no mutations predicted as leading to TMP resistance are colored in white. Positions with at least one mutation predicted as leading to TMP resistance with over 0.5 (classification threshold, 28 positions) are colored in light blue. Positions with at least one mutation predicted as leading to TMP resistance with over 0.8 (high confidence predictions, 14 positions) are colored in dark blue. All positions with at least one mutation with threshold ≥ 0.8 are within 11.5 Å of at least one TMP heavy atom.

of -0.233), F199D (probability of 0.61, MTX of 0.673, 37°C of -0.242) and F199G (probability of 0.61, MTX of 0.638, 37°C of -0.335) are part of the 11 mutations where a tradeoff between MTX resistance and function was observed.

The model also predicted several mutations that did not lead to MTX resistance. All mutations predicted as leading to resistance are in the right range of distance to ligand and buriedness. These mutations, I10K (probability of 0.94), F36C (0.91), F121H (0.52), I123G (0.82), F156V (0.75) and Y197Q (0.81) highlight the model's ability to make novel predictions based on ligand-dependent features, which vary between the PjDHFR-MTX and PjDHFR-TMP structures (S6B Fig). Of these, F36C has been observed in the clinic and biochemically characterized as reducing affinity between PjDHFR and TMP. Despite the importance of ligand-independent variables, such as buriedness and wild-type side chain length, the presented model can differentiate and make predictions based on changes in bound molecules, and this without explicit structural information. All data used to make predictions on the effect of mutations on resistance to TMP is available in S2 File.

Next, we examined if our predictions captured mutations previously shown or suspected to cause resistance, for instance ones that have been identified in clinical studies and/or biochemically characterized. Few are available, and fewer yet have been linked to failure of treatment, biochemically characterized, or identified in more than one study (S2 Table) [22,24,25,46–48]. Amongst the mutations identified in patient samples that have been characterized *in vitro*, three had been identified as being most likely resistance-causing, based on their impact on $K_i$, $K_m$ and $k_{cat}$ [22,24]. These amino acid changes, F36C, L65P, and A67V, caused 100-fold, 20-fold and 40-fold increases in TMP $K_i$ in PjDHFR, respectively. The classification model predicted mutation F36C as leading to TMP resistance with a confidence of 0.91 but did not predict mutations L65P (Resistance probability 0.33) and A67V (0.03) as being likely to lead to resistance (S7C fig for details on model decision). While all three mutations have effects on biochemical constants that could be linked to resistance, their effect on function had not been investigated in a biological system prior to this study. We found that several mutations at position L65, including L65P, caused SDE, implying that this mutation cannot lead to TMP resistance *in vivo*. However, while L65P was not identified as being likely to lead to TMP resistance, mutation L65G was, with a prediction of 0.58, which is coherent with the large effect on $K_i$ caused by another mutation at this position. Neither F36C nor L65G were experimentally identified as leading to MTX resistance in our previous study, meaning that the model can make novel predictions based on data it had not encountered previously.

**PjDHFRs from clinical samples show no mutations associated with increased organismal load in patients or predicted as leading to TMP resistance**

Despite increased interest on the effect of mutations in PjDHFR on treatment efficiency and patient outcome, the size of the available dataset on sequence polymorphism is limited. To try and expand this dataset, and screen for potential informative resistance mutations from patient samples, we sequenced PjDHFR in *Pneumocystis jirovecii*-positive samples from the Laboratoire de Santé Publique du Québec (LSPQ), a large public laboratory in Eastern Canada. We used multiplexed amplicon sequencing specifically targeting PjDHFR in these samples. Using data from a standard qPCR assay able to detect intervals between 3 000 and 3 000 000 genomic copies per mL of bronchoalveolar wash, we investigated if DHFR mutations correlate with larger organismal load in patients.

Between 2019–2024, 636 respiratory samples were found positive for *P. jirovecii* as part of the *Pneumocystis* qPCR detection and quantification program (>5 000 genome copies/mL) from across the province of Quebec. From these, we successfully amplified and sequenced the coding sequence of 531 PjDHFRs. As samples were sequenced with a minimum of 25x coverage across all bases, we identified the presence of different alleles within individual samples, when polymorphisms were identified in at least 30% of reads at a given position. We identified 104 samples (19.6%) containing substitutions relative to the *P. jirovecii* reference genome (RU7), with 94 harboring synonymous substitutions (17.7%). The mutations found most often were the previously reported G90G (T270C in the spliced sequence) silent mutation [25–27,49], as well as novel synonymous ones (S11A Fig). Ten samples had amino acid substitutions, including the previously reported

G18W, A67V and C166Y, and 2 nonsense mutations, L105* and E149* [22,25,26]. A67V was one of two amino acid changes identified in more than one sample, being found in four samples isolated at different hospitals between 2020 and 2024, and C166Y, found in two samples. A subset of eight samples was validated by Sanger sequencing, and all matched the amplicon sequencing results. We identified 44 samples (8.2% of all samples, 42.3% of samples with at least one polymorphism) that likely had two different PjDHFR alleles (S11B Fig). When found, mutation A67V was always accompanied by a wild-type allele. As *P. jirovecii* is suspected to produce a diploid zygote as part of its mainly homothallic sexual reproduction, we cannot confidently say that the presence of two alleles is due to coinfection [50,51]. However, as the transmissible form of *P. jirovecii* is suspected to be haploid, coupled with the 25x read threshold we set for both assembly and polymorphisms, we believe that co-infection by two different strains is a likely cause of the presence of different alleles within single samples, as shown by others [14,51,52]. Most importantly, of all the mutations identified from clinical samples in this study, none were predicted by the model to lead to TMP resistance in PjDHFR.

Samples were classified as being either wild-type, containing synonymous, non-synonymous or nonsense mutations. We then compared each class with the qPCR count of *P. jirovecii* genome copies within samples, and copy counts were standardized to account for the method's detection limit of between 3 000 and 3 000 000 copies. We found that the presence of non-synonymous mutations did not lead to a statistically significantly higher qPCR count than in samples with no sequence changes relative to the reference (S9C Fig). For substitution A67V, three out of four samples had counts of less than 20 000, less than half of the median across all samples (~43 000 copies). The fourth sample containing this mutation had a qPCR count of 1 794 000 copies, being above the 85th percentile of the distribution. From this distribution of counts, we cannot confidently associate A67V with a higher *P. jirovecii* load in patients, despite biochemical evidence that this mutation reduces affinity between TMP and PjDHFR. These results, where mutations in PjDHFR are not associated with higher organism load, are coherent with the presented model not predicting the identified mutations as being resistance-causing. Furthermore, across all 531 sequenced samples, 265 also had their DHPS sequenced during multi-locus sequence typing at the LSPQ, and mutations were only identified in three samples, with only two being at position 55, associated in DHPS with SMX resistance. We find no overlap between model predictions of TMP resistance and mutations found in PjDHFR from patient samples. In addition, we find no correlation between the presence of mutations and organismal load in patients' samples, and therefore can make no associations between mutations in PjDHFR and infection, treatment and outcome.

## Discussion

### A combined experimental and computational framework for making predictions on antifolate resistance in *P. jirovecii's* DHFR

The study of how mutations in *P. jirovecii* affect its sensitivity to antifolate drugs has always been difficult. As TMP and MTX share structural and mechanistic properties, we hypothesized that data obtained on the effects of mutations in PjDHFR on resistance to MTX could help predict the effect of mutations on resistance to TMP (Fig 1). Importantly, we framed this hypothesis as a method to investigate cross-resistance and transferability rather than an assumption of equivalence between the two molecules, as even closely related antifolates may differ in subtle but biologically meaningful ways. From this, we hoped to shed light on both ligand-dependent and independent features leading to sensitivity or resistance.

By using experimental data from DMS screens on resistance to MTX and on protein function, and computational data from various modeling strategies, we trained a machine learning model to predict the effects of mutations in PjDHFR on resistance to TMP (Figs 2 and 3). By focusing on an interpretable machine learning framework, we aimed to present an integrative approach to combine heterogeneous functional, structural, and evolutionary features that interact, thereby allowing mechanistic insight rather than purely predictive output. The resulting random forest model accurately captures the mutually exclusive character of SDE mutations and resistance, as well as the effects of modeled features such as

protein/ligand contacts, changes in protein free folding energy, and residue conservation. While there was a significant overlap between the predictions made on the effects of amino acid substitutions on resistance to MTX and TMP, the model predicts substitutions that were not classified as MTX resistant would lead to TMP resistance, highlighting its ability to make novel predictions based on new data (Fig 4). These discrepancies underscore that resistance determinants may partially overlap, but are not fully interchangeable across antifolates, reinforcing the need to use integrative modeling when predicting the effects of mutations in drug resistance. Furthermore, despite having overlapping binding moieties, the structural differences between MTX and TMP possibly prevent the model from accurately predicting the effects of mutations which solely interact with the diverging regions of the drugs.

No single modeling method can capture all effects of amino acid substitutions in PjDHFR, as each focuses on individual aspects of protein properties. Different positions have different levels of stringency for specific amino acid properties. Therefore, prediction of mutational effects should be conducted using several tools focusing on different features, and proteins must be investigated in a per-residue manner to identify tolerated permutations at given positions. Other properties, such as which residues are involved in catalysis and which residues are allosterically linked, often still need to be experimentally determined [44,45,53]. While computational methods can allow the modeling of granular properties (FlexddG, MutateX) or integrative scores (GEMME), certain experimental methods cannot yet be fully replaced, and are still necessary to assess biological functions. Particularly, our results demonstrate that experimentally measured functional landscapes derived from DMS provide critical constraints that substantially enhance predictive performance compared to computational descriptors alone.

Of the three mutations of interest identified in clinical samples that had been biochemically characterized in [24], the model predicted that one would lead to TMP resistance (F36C, not identified as leading to high levels of MTX resistance), one was predicted as not leading to TMP resistance, as it was experimentally identified as non-functional (L65P), and one was predicted as not leading to TMP resistance because of other factors (A67V). To further compare model predictions to clinical data, we sequenced PjDHFRs from 531 patient samples from across the province of Québec and identified both new and previously reported PjDHFR polymorphisms. No mutations identified from these patient samples were predicted as leading to TMP resistance, and none was consistently associated with high organismal load. While these observations do not constitute functional validation of resistance, they are consistent with the relatively low predicted number of high-confidence resistance-causing mutations predicted by the model. This sequencing experiment also highlighted endemic sequence variation in PjDHFR sequence, as well as potential dynamics of coinfection by different *P. jirovecii* strains within patients.

As there yet exists no experimental methods to validate the predictions made by the presented model, or to test mutants identified from clinical samples for resistance to TMP, the presented results remain predictive. We therefore interpret TMP resistance predictions probabilistically, particularly for variants with prediction scores near the classification threshold. Despite this, our study offers novel resources to help understand mechanisms of antifolate resistance in PjDHFR, as well as a general workflow to predict the effect of mutations on resistance in other hard-to-study organisms. By identifying which features correlate best with MTX resistance and building a predictive model which we could validate using DMS data on MTX resistance, we believe we were able to make informed predictions about the effects of those same mutations on TMP resistance. With a balanced accuracy score of 88.3% in the testing set for MTX resistance, we expect that predictions regarding the effect of mutations on TMP resistance should have a similar overall balanced accuracy. Nevertheless, differences in molecular interactions between MTX and TMP may limit full transferability, and experimental validation of high-confidence TMP predictions will be required to establish their clinical relevance.

We believe that this framework could be applied to other protein/ligand combinations for which DMS data is available, and potentially help orient drug development. Indeed, by identifying position-dependent resistance-causing features, therapeutic molecules could be tailored to maximize binding to mutants that cause resistance to other therapeutic agents. Furthermore, this framework could help direct experimental testing efforts to specific positions of interest across

the protein. These positions could be investigated individually, without the need for costly full-protein DMS, where it is expected that most mutations will have a neutral or deleterious effect. With the increase in available DMS data for many clinically relevant proteins/ligand pairs, such frameworks are critical to make the most of the large dataset generated by these experiments.

## Functional data from DMS screens reveals widespread lack of tradeoff in antifolate resistance and enhances TMP resistance model accuracy

When conducting a DMS screen on the effects of PjDHFR mutations on protein function, we observed that 14.2% of mutations lead to an SDE phenotype, which is consistent with reports in other essential fungal proteins and in bacterial DHFR [40,43,54]. Such mutations were concentrated at positions with very low tolerance, where most mutations lead to SDE, such as the 9 Å diameter region of PjDHFR with a median selection coefficient of -0.363 (S4 Fig). Often, when only a subset of mutations led to SDE at a given position, these mutations shared functional characteristics such as side-chain length, polarity, or charge. An exception were substitutions towards proline, which were consistently more deleterious across the protein, consistent with previous reports [38,39,55]. Very few mutations were measured as having an intermediate effect on function (58/3758, 1.54%). This could be explained by the relatively high strength promoter used in the system, as high expression can mask the effects of slightly destabilizing mutations [56]. Consistent with the small number of intermediate-effect mutations, even fewer mutations (11/3758, 0.29%) exhibited a tradeoff between function and MTX resistance (Fig 2C). This limited trade-off highlights a strong functional constraint: resistance-conferring mutations must preserve sufficient enzymatic activity, reinforcing the importance of directly measuring mutant functionality when modeling resistance. Otherwise, such critical effects may be omitted by predicting algorithms.

Experimental measurement of mutant functionality was critical to model performance. This feature ranked as the second most important predictive one and helped minimize both false positives and false negatives in the random forest classification model (Figs 3 and S8). In most cases, resistance and SDE were mutually exclusive, enabling accurate classification of non-functional mutants as non-resistant. Furthermore, several top predictive features were at least partially correlated with functionality. Having direct experimental data on this phenotype proved essential for training reliable classification models. This model, when trained without functional DMS measurements, showed reduced discriminatory power, highlighting the key importance of experimental functional constraints over purely computational predictors. While this can represent a major hurdle when working on hard-to-study organisms such as *P. jirovecii*, the importance of this feature on model performance suggests that integrating experimentally measured functional data may be broadly necessary for reliable prediction of drug resistance in other systems as well.

Furthermore, having experimental data on the effect of mutations on protein function was key to minimize false positives, when compared to other models [28]. Overall, the best-performing features included ligand-independent features such as mutant functionality, residue buriedness, wild-type side chain length, and predicted changes in protein free energy (MutateX). Critical ligand-dependent features include proximity to ligand heavy atoms and changes in region flexibility, modeled via FlexddG (Fig 3A, S1 Table).

## Class imbalance negatively affect model performance when predicting antifungal resistance

When working with antifungal resistance, a minority of all possible single mutations will lead to resistance, unless SDE is enough by itself to cause resistance [54]. While it is advantageous from a clinical standpoint, it can cause ML datasets classes (sensitive/resistant) to be highly unbalanced, with a large majority of mutations having little to no effect on resistance and can therefore lead to poor model fitting. We had previously established a method that uses a Gaussian mixture model to dissect complex distributions, which allowed us to establish thresholds for MTX resistance and function (S4 Fig). Only 111/3758 (2.8%/97.2%) mutations in PjDHFR were identified as leading to MTX resistance, which leads to imbalanced class distribution in the dataset, and therefore a potentially limited predictive power. Despite this, when

training predictive models such as this one, we believe that it is critical to have data on as many amino acid substitutions as possible, hence the relevance of both experimental and *in silico* DMS. Having an interpretable model also allows us to rationalize the mechanisms of resistance, which then allows us to use critical judgment on the predictions that lie near the threshold used for binary classification based on the nature of the mutations involved.

While the model accurately classified the majority of mutations, this imbalance in classes is reflected in some evaluation metrics. For instance, the area under the curve (AUC) of the receiver operator characteristic curve (ROC), not affected by dataset imbalance, being high (0.97 on the test set) is explained by the very high amount of true negatives and indicates a well performing model. However, the AUC of the precision-recall display curve (PRD), which is much more sensitive to class imbalance, being lower (0.41 on the test set) is caused by the relatively high number of false positives and low number of true positives, which negatively impacts precision (Fig 3A). True negatives, while influencing the ROC curve, do not affect the PRD curve. It is key to use several evaluation metrics when training and testing classification models, and be critical of model performance, based on desired outcomes. Furthermore, when working with datasets with such imbalance, bootstrapping, oversampling and SMOTE sampling often lead to overfitting, minimizing predictive power outside of the training dataset, and should therefore be avoided when possible [57]. This was also reflected in the balanced accuracy scores obtained for training (98.3%) and testing (88.3%). With 21 possible true positives in the test set (2.79% of observations), misclassification of mutants as either false negative or false positive has a large impact on balanced accuracy. Class imbalance in relatively small datasets such as this one (< 5000 observations) has been known to affect prediction accuracy.

Binary classification required the definition of a probability threshold. Here, we applied the conventional cutoff of 0.5. However, such a threshold does not directly reflect the rate (2.8%) of resistant mutants in the dataset. In this context, we consider that the 0.5 threshold represents a neutral and conservative threshold rather than a biologically relevant boundary, as opposed to the experimentally measured resistance threshold. Biological interpretation should rely primarily on continuous probability outputs and feature importance analyses rather than strict classification of variants, especially near the decision boundary.

While the ML-based methodology presented in this study uses experimental measurements for model training and validation, other variant effects predictor software use purely computational features for ML-less modeling and prediction [58,59]. One such tool is Resistor. Based on the protein design software Osprey, this algorithm uses predicted changes in binding affinity to ligands caused by mutations as well as their synergistic effects and makes ranked predictions based on mutation likelihood. Such algorithms are not affected by dataset class balance as they do not require *ab initio* ground truths like ML-based models. They also focus on fewer, empirical features, and therefore offer a complimentary approach to ML-based methods such as the one presented here [58,59].

Nevertheless, the model successfully predicted that certain mutations, not measured as conferring MTX resistance, would lead to TMP resistance in PjDHFR. This included mutations found in literature and biochemically identified as reducing affinity to TMP, with one of the main predictors, region flexibility (here IQR change in distance) being identified as a good predictor of TMP resistance in a previous study [28].

## Expanding the clinical genomics landscape of *P. jirovecii* DHFR highlight endemic genetic variation

To illustrate how our model could be put to use, we sequenced the largest dataset of PjDHFR from clinical isolates presented in literature to date and compounded the results of many previous studies. Despite sequencing PjDHFRs from 531 patient samples spread across five years, different geographical areas, with differences in organismal loads, and finding both new and previously reported mutations, no mutations from this patient cohort could be confidently linked to TMP-SMX resistance. However, we believe that this high-throughput sequencing of PjDHFRs sheds light on endemically circulating genotypes of PjDHFRs in eastern Canada, such as G90G, A67V and C166Y, as well as co-infection by different strains of *P. jirovecii*. While the DHPS has been sequenced for many clinical samples when PCP was diagnosed, the

PLOS Genetics

DHFR locus had not been as thoroughly investigated in this organism before this study. The lack of genetic information on the endemic and clinical genetic variation in PjDHFR compounds the difficulty to study resistance on this system. There is a need to sequence further genomes of *P. jirovecii*, or at least of its drug targets, even in the event of treatment success. To further validate these results, it is necessary to conduct such high-throughput sequencing studies across other geographical regions to uncover more information on the evolution of TMP-SMX resistance/treatment failure in *P. jirovecii,* including in other antifolate-resistance associated genes with robust clinical outcome data. This would be especially relevant in regions of the world where failure of prophylaxis and treatment is high, and where second-line treatments are less available [60]. As the mechanism of action of TMP-SMX in this fungi is still poorly understood, with antifungal activity often solely attributed to SMX, such studies could help better understand how this drug combination contributes to treatment outcome [61,62].

## Conclusion

Recent advances in experimental technologies enable an unprecedented level of granularity in studying the effects of mutations on proteins. However, applying these high-throughput methods to hard-to-study organisms, such as unculturable organisms, remains challenging. In this study, we show that experimentally derived functional landscapes can partially overcome this limitation by revealing the biological constraints that shape the evolution of drug resistance. Computational methods and machine learning are key tools to address these limitations but are not yet accurate enough to replace experimental validation, particularly when addressing complex phenomena like the evolution of resistance. Rather than serving solely as predictive engines, such approaches are most informative when integrating functional, structural, and evolutionary features, and exposing trade-offs between protein function and drug resistance, as well as non-linear feature interactions within proteins. Furthermore, if the experimental investigation of resistance to TMP is ever possible in PjDHFR, feature importance between resistance to MTX and TMP could help highlight the determinants of cross-resistance.

Overall, we believe that high-throughput computational methods, combined with the pattern-recognition capabilities of machine learning, are critical for guiding experimental design and identification of key determinants of drug resistance, especially in hard-to-study organisms. Our results highlight that resistance-conferring mutations are constrained by the need to preserve essential enzyme function, alongside which features correlate best with this, providing insight into the predictability and limits of resistance evolution in PjDHFR. More broadly, this framework illustrates how resistance mechanisms learned for one drug can inform, but not fully determine, resistance to related compounds, with implications for antifolate drug design, resistance surveillance, and the study of evolutionary dynamics in unculturable pathogens.

## Materials & Methods

### General methods

All yeast transformations were conducted using the standard lithium acetate transformation protocol with heat-shock temperature adjusted to 25°C when using temperature sensitive strains [63]. All PCR reactions were conducted using KAPA HiFi Hotstart DNA polymerase (Sigma-Aldrich, catalog #BK1000) unless stated otherwise. All mutations identified from the literature and from clinical samples in this study are provided in S2 Table. Details for all strains are available in S3 Table, plasmids in S4 Table, oligonucleotides in S5 Table, media in S6 Table and PCR reactions in S7 Table.

### Temperature sensitive DHFR strains

Yeast strains used in this study were taken from the Yeast Temperature-sensitive (TS) collection [64]. TS-DHFR replicates were taken from individual wells from the collection as biological replicates, and streaked on YPD + 200μg/mL Geneticin (YPD + G418) agar plates (Plate 4A position B8, plate 4B position B7, plate 4C position A8, plate 4D position A7 in the TS collection [64]). Four isolated colonies were harvested from each biological replicate and grown in YPD + G418 liquid

media. Each culture was diluted to 1.0 $OD_{600}$ and serial-dilution spot assays were conducted on YPD + G418 plates and grown at 20°C and 37°C for 72h with pictures taken every 24h (S1 Fig). A colony from each biological replicate that exhibited growth at 20°C but not at 37°C (Colony #1, S1 Fig) was selected to ensure the presence of the temperature-sensitive phenotype. Complementation by PjDHFR was tested by transforming pAG416-GPD-PjDHFR into all four strains and observing growth at 37°C in synthetic-complete solid media without uracil and with monosodium glutamate (SC-URA + MSG) media.

All subsequent experiments were conducted in quadruplicate using each individual TS-DHFR strain. All subsequent experiments with these strains were conducted at 20°C unless stated otherwise.

**PjDHFR DMS library transformation and pooled screens**

The PjDHFR DMS library from [36] was pooled in equivalent molar concentrations by position in four fragments, with an 11 codon overlap between each adjacent fragment. Fragment 1 comprised positions 2–63, fragment 2 from 52 to 115, fragment 3 from 104 to 167 and fragment 4 from 156 to 206. Each fragment pool was individually transformed into competent cells from each of the four TS-DHFR strains (biological replicates), with 16 technical replicates per TS-DHFR strain, and incubated at 20°C for seven days in SC-URA + MSG media to minimize the selection bias against slow-growing PjDHFR mutants. Using SC-URA + MSG liquid media, the colonies on plates were harvested in sterile falcon tubes for each fragment and biological replicates, with technical replicates pooled at this step. These were diluted to 0.1 $OD_{600}$ in fresh liquid SC-URA + MSG and grown at both 20°C and 37°C with 250 RPM orbital agitation for 24h, then passed to 0.1 $OD_{600}$ in fresh liquid SC-URA + MSG for another 24h. At each timepoint, 5.0 $OD_{600}$ units were spun down at 224 g for 3 minutes, and pellets were harvested for downstream DNA extraction and frozen at -80°C. Glycerol stocks were made in triplicate.

Plasmid DNA extractions were conducted using Zymoprep Yeast Plasmid Miniprep II (Zymo Research, cat #D2004), with an added pellet freezing step after initial pelleting, and with 3h incubation with zymolyase. Extracted plasmids were amplified using the primers listed in S5 Table and purified using using magnetic beads (AMPureXP from Beckman Coulter Life Sciences, catalog #A63882). Sequencing was conducted using the Element AVITI with paired-end 150 bp, yielding ~100M total reads, resulting in 1800 reads per variant at each timepoint and condition. Parsed and analysed mutation calling is available in S4 File for the pooled assay at 20°C (permissive condition) and S5 File for the pooled assay at 37°C (selective condition). Raw reads for the function screen (both temperatures) are available at BioProject PRJNA1243435.

**Read processing, mutations calling and statistical analysis**

Reads were processed and analyzed as described in [36]. Briefly, R1 and R2 reads were merged using Pandaseq (v2.11) using default PHRED score threshold [65], and aggregated and counted using VSEARCH (v2.22.1) [66]. Changes in read frequency were calculated between $T_0$ and $T_{Final}$ for all variants. Frequencies were calculated as the proportion of reads containing a given codon relative to the total number of reads in that sample. Selection coefficients for all variants were computed using the following equation, with the median score of silent mutations relative to the WT sequence as the normalization baseline (wild-type score):

$$Selection\ coefficient\ =\ \frac{Log2(\frac{Freq\ mutant\ T_{final}}{Median\ freq\ silent\ T_{final}})\ -\ Log2(\frac{Freq\ mutant\ T_0}{Median\ freq\ silent\ T_0})}{g}$$

(1)

Where g, the number of generations, was estimated based on optical density measured at the start and end of the different timepoints. Mutations in codons that lead to the same amino acid substitutions were grouped after computing the selection coefficient, and the median score was used for subsequent analysis of the effect of amino acid substitution. Statistical analysis, thresholding and control for false-discovery rate were conducted in an identical manner as described

in [36]. Using the scikit-learn (sklearn, v1.5.1) GaussianMixture algorithm, the distributions of fitness scores were decomposed in their underlying Gaussian distributions, reflecting the statistical distribution of selection coefficients, and thresholds were set accordingly. We controlled for false discovery rate using the distribution of silent mutations as reference for a Welch's one-sided *t*-test followed by a Benjamini-Hochberg (BH) FDR control with a threshold of 0.05 (S4 Fig).

## Structural analysis modeling of changes in protein free energy and in binding affinity to ligands

The structure of PjDHFR was recovered from AlphaFoldDB (UniProt: A0EPZ9). The crystal structures of DHFR, in complex with MTX/NADPH (PDB: 3 CD2) DHF/NADPH (PDB: 4 CD2), and TMP/NADPH (PDB: 1DYR) were recovered from the Protein Data Bank (PDB). Using the ChimeraX (version 1.6) [67] matchmaker algorithm, the structures were aligned with each other and ligands from crystal structures were extracted and added to the PjDHFR AlphaFold structure.

The resulting ligand-protein complex structures were then relaxed ten times using the FoldX (suite 5) optimize command to remove van der Waals clashes [68]. We used MutateX v0.8 [33] protocol from FoldX suite 5 [68] on the entire protein, with 10 incorporated minimization steps following mutations to reduce atom clashes sometimes caused by longer sidechains, before ΔΔG measurements [69]. Apart from additional minimization steps, analysis was conducted as described in [36].

Using RosettaDDGPrediction (Rosetta v3.12 and 2022.11), adapted with custom scripts to allow high-throughput computing, the FlexddG protocol with saturation mutagenesis was run across the entire length of the PjDHFR-ligand structures [34, 35, 70]. The effects of all mutations were measured on the changes in Gibbs free energy (ΔΔG) between PjDHFR and the ligand of interest, either MTX, TMP or NADPH. This was done 100 times with 35 000 backrub steps for each mutant and all individual structures were analyzed with a modified version of the basic FlexddG analysis script, available on GitHub [34]. As final scores, the median ΔΔG of each mutation was used for downstream analysis.

We measured the changes in distance between the α-carbon of the mutated and wild-type residue and the closest heavy atom on the ligand. FlexddG generates, for each mutation of interest, a structure for both the wild-type and the mutated sequence before backrub, and measures ΔΔG by comparing the wild-type structure's ΔG with the mutant's ΔG. By using each structure, we could measure the effect of each mutation on the protein's flexibility by comparing the position of the α-carbon in the wild-type and mutant structures, which yielded the root-mean-square fluctuation of each structure, or the change in distance to ligand between the wild-type structure and the mutant. This allowed us to get a distribution of the effective structural variation caused by the mutation. From this distribution, we extracted three values: 1) the median value of changes in distance, representing the overall change in residue position caused by the mutation, 2) the skewness of the distribution, which represents the asymmetry of the distribution, or the region's propensity to be in a region of space more than in another, either generally closer or further from the ligand, and 3) the interquartile range of the distribution, IQR, representing the "flexibility" of the protein region caused by the mutation. A wide distribution (high IQR) means the mutated residue increases region flexibility, as the mutated region can occupy a broader range of positions further from its wild-type position.

## Sequence conservation and epistatic interaction between positions

Using the MPI Bioinformatics Toolkit portal, we ran HHblits on the reference RU7 PjDHFR amino acid sequence using default settings [49,71,72]. The resulting 1040 sequences with < 70% sequence homology were aligned, and gaps were removed (S5 File, raw alignment in S6 File). This alignment was used as the input for GEMME to compute epistasis and conservation effects scores [37]. All three scores, Combined, Epistasis and Individual, were compiled for all possible single mutants based on sequence conservation and co-occurrence across lineages. All data for GEMME analysis, and its output, is available on GitHub.

## Other amino acid and position properties

Amino acid properties for both wild-type and mutant sequences were defined using Expasy amino acid properties [73]. All available properties were correlated with each other, and property with the highest median correlation within each group was kept to minimize the dimensionality of the dataset. Variance inflation factor (VIF) was also used to minimize

correlations within the dataset. Buriedness was computed using a script available at https://github.com/rodogi/buriedness. git, which is a metric similar to accessible surface area (ASA), but with no zero values, allowing comparison between buried and very buried residues, being more informative about relative residue depth than strict ASA.

The same script used to measure changes in position in FlexddG structures was used to measure distances between ligands and α-carbon in the wild-type protein as the distance to ligand feature. We used the DSSP algorithm to assign letter scores to all positions to identify if they are part of secondary structures [74, 75].

### Machine learning mo del selection, training, and testing

To predict the effect of mutations on PjDHFR's resistance to TMP, the PjDHFR-TMP dataset (S2 File) was used. The features used to train the model were FlexddG score, MutateX score, a binary classification of the mutant's functionality, α- carbon distance to the closest ligand's heavy atom, GEMME scores, and effects of the mutation on structure flexibility, as measured from FlexddG output structures, the length of the side chain of the wild-type amino acid, allostery confidence score, importance in catalysis and buriedness were also used (S1 Table) [40,44,76]. Other features present in the raw dataframe were dropped from the dataset before training but are kept in the raw file to help visualise model performance after training and testing.

The model was created using scikit-learn (scikit-learn, v1.5.1) [77]. Using a fixed seed (999), the dataset was split into training and test sets, with 20% reserved for testing, using train_test_split(). Data was scaled using the standard scaler provided by the package, StandardScaler(), which performs z-score standardisation, helpful to make all data on similar scales [77]. Both classifier and regression methods were tested, including but not limited to logistic regression, random forests, decision trees, support vector machines and XGboost. A randomized hyperparameter search was used for broad model selection, estimating performance using the median across five cross-validation. Across all models, the random forest method consistently performed best. Classification performance was assessed using precision, recall, F1-score, and log loss (binary cross-entropy), while regression models were evaluated using RMSE, $R^2$, and rank correlation.

Once the best performing method was selected, an exhaustive grid search was performed, with parameters granularity defined according to the computation time required by the given model. We used SHapley Additive exPlanations (SHAP, v0.42.1) values to identify which model features could best explain the effect of mutations on resistance to MTX [78].

### Predictions for the effect of mutations on TMP resistance in PjDHFR

In parallel with generating data on the PjDHFR-MTX complex, we performed the same analyses on the PjDHFR-TMP complex and compiled a dataset using the same features as those used to train the ML model. This dataset was then used to predict the impact of mutations on TMP resistance using the trained model. Using the predict(), and predict_ proba() methods from the sklearn package, predictions on TMP resistance-causing mutations were made from the dataset generated with the PjDHFR-TMP complex. All code used to train the model on MTX resistance, and to make predictions on TMP resistance, is available on GitHub.

### Sequencing PjDHFR from clinical samples

Primary samples were acquired by the Laboratoire de Santé Publique du Québec, in Montréal, as part of a province-wide *Pneumocystis jirovecii* monitoring program. In total, 636 samples were obtained from December 2018 to May 2024 from 41 hospitals across the province. DNA was extracted using Biomerieux's EMAG standard protocol. qPCR was conducted using Altona Diagnostics' RealStar *Pneumocystis jirovecii* PCR Kit 1.0 (Cat #551013) to quantify *Pneumocystis jirovecii* genomic copy numbers from standardized primary sample volumes. For this specific kit, no housekeeping gene is used as a copy number normalization control, but samples are systematically tested for the presence of PCR inhibitors to minimize copy count bias.

Primers specifically targeting PjDHFR were then used on the extracted DNA to obtain a full length PjDHFR amplicon [79]. We used the Row-Column-Plate PCR method described in [80], with further steps to minimize inter-sample contamination. Briefly, full length PjDHFRs amplicons were split into 3 fragments using fragment-specific primers (F1 - 586 bp, F2 - 579 bp and F3 - 533 bp, 113 bp overlap between adjacent fragments), and barcoded in a sample-wise manner, where fragments from the same sample shared barcodes. In total, 531 PjDHFR amplicons were prepared for pooled amplicon sequencing using Illumina MiSeq for ~700 000 PE300 reads, or ~1300 reads per PjDHFR sample (Oligos in S4 Table).

Using custom scripts, sequencing data was parsed and assembled into full length sequences (bwa-mem2 v2.2.1, samtools v1.21, cutadapt v2.6 [81–83].) To ensure high-quality assembly, samples with fewer than 125 total reads or fewer than 25 reads per fragment were excluded. After filtering, 531 high-quality PjDHFR sequences were successfully assembled from clinical samples. Using custom scripts, SNPs were called by comparing assembled amplicons to the RU7 reference sequence. With a minimum of 25x coverage at every position, we were able to identify intra-sample polymorphisms. Polymorphisms were called when a non-consensus base was present in over 30% of reads mapping to a given position. All raw reads for amplicon sequencing of clinical samples are available at BioProject SAMN47937957.

## Supporting information

**S1 File. MTX training and testing dataset with model predictions.** Contains dataset for training and testing the machine learning model, as well as prediction outputs from the model for PjDHFR-MTX.
(CSV)

**S2 File. TMP dataset with model predictions.** Contains dataset for making predictions on PjDHFR-TMP, as well as prediction output for PjDHFR-TMP. Data is not de-scaled.
(CSV)

**S3 File. DMS screening data of the function assay in permissive condition.** Parsed read data in long format for screening done at 20°C for all fragments in all strains, in codon format.
(CSV)

**S4 File. DMS screening data of the function assay in selective condition.** Parsed read data in long format for screening done at 37°C for all fragments in all strains, in codon format.
(CSV)

**S5 File. DHFRs aligned using HHblits with gaps removed.** HHBlits DHFR alignment with gaps removed for GEMME analysis. All DHFRs have <70% homology to PjDHFR to not skew GEMME output with highly similar sequences.
(TXT)

**S6 File. DHFR aligned using HHblits with gaps presents.** HHBlits DHFR alignment with gaps present. All DHFRs have <70% homology to PjDHFR to not skew GEMME output with highly similar sequences.
(FASTA)

**S1 Table. Description and reference for all features used in model training.**
(XLSX)

**S2 Table. Compilation of DHFR mutations leading to drug resistance found in literature, as well as mutations found in this study.**
(XLSX)

**S3 Table. Details on strains used in this study.**
(XLSX)

**S4 Table. Details on plasmids used in this study.**
(XLSX)

**S5 Table. Details on oligonucleotides used in this study.**
(XLSX)

**S6 Table. Details on media used in this study.**
(XLSX)

**S7 Table. Details on PCR cycles used in this study.**
(XLSX)

**S1 Fig. Spot-dilution assay to validate temperature-dependent growth of the four TS-DHFR strains.** From each TS-DHFR strain from the temperature sensitive allele collection [64], samples from the collection glycerol stock were spread on YPD+G418 plates. From each biological replicate, four isolated colonies were selected and grown overnight in YPD+G418 liquid media. Cultures were then diluted to 1.0 $OD_{600}$ and spotted on YPD+G418 plates with 5-fold serial dilutions. One plate was incubated at 20°C (permissive condition, top) and at 37°C (selection condition, bottom) for 72h. On each plate, strain IGA130 (*DHFR*-WT) and strain FDR0001 (Δ*dhfr,* inducible DfrB1 to allow for growth) were used as positive and negative growth controls, respectively [36].
(TIFF)

**S2 Fig. Correlation among biological replicates of screens at 20°C and 37°C.** A) Scatterplots of selection coefficients for mutated codons in the different biological replicates at 20°C. Axes are scaled to allow comparison of the different conditions. Density plots show the distribution of selection coefficients for the replicate on the x-axis. B) Scatterplots of selection coefficients for mutated codons in the different biological replicates at 37°C. Axes are scaled to allow comparison of the different conditions. C) Scatterplots of selection coefficients for amino acids present in the overlap between the different fragments at 20°C to ensure good correlation between the different fragments. D) Scatterplots of selection coefficients for amino acids present in the overlap between the different fragments at 37°C to ensure good correlation between the different fragments. Scatterplots for each fragment overlap show similar patterns at 20°C and at 37°C.
(TIFF)

**S3 Fig. Heatmap of selection coefficients for screening done at 20°C.** Selection coefficient is measured relative to the median of silent mutations. Positions of contacting residues along PjDHFR (PDB: 3 CD2 (MTX and NADPH), 4 CD2 (DHF) and 1DYR (TMP)). Contact was established as amino acids with an α-carbon located less than 8 Å from MTX, DHF, TMP or NADPH. Black dots represent the wild-type sequence.
(TIFF)

**S4 Fig. Gaussian mixture model optimization for the permissive condition (20°C) and non-permissive condition (37°C).** Optimization of information criteria for A) 20°C and B) 37°C. Best Gaussian mixture model (dashed lines represent underlying Gaussians) to recapitulate the underlying distribution (black line/ histogram) of C) 20°C and D) 37°C. Density curve of significant mutants for Benjamini-Hochberg (green) correction is visible. E) Intersection between statistically significant mutants in 20°C and 37°C. Upset plot showing the intersections of the Benjamini Hochberg-FDR corrected groups (control for the false discovery rate of significantly resistant mutants at 95% confidence).
(TIFF)

**S5 Fig. Mutation effect on function at 37°C mapped on structure and analysis of region with a proportionally higher rate of SDE.** A) Position-wise minimum score, colored as in Fig 2A. B) Position-wise median score. C) Position-wise maximum score. D) Visualization of the region with a high rate of SDE. Positions 54–64 are colored in blue,

positions 75–81 in yellow and 121–126 in red. E) Distribution of GEMMECombined scores for all positions in grey, positions 54–64 are colored in blue, positions 75–81 in yellow and 121–126 in red. One-sided Mann-Whitney U scores and p-values for each distribution are shown. Mutations in these regions are predicted as being more likely to be deleterious than most mutations across the protein. NADPH is colored in green, and folate is colored in black.
(TIFF)

**S6 Fig. Variance in experimentally measured functional scores explained by individual computational features.** Distribution of feature values are normalized between 0 and 1 for each feature used in the analysis, separated by variants classified as functional or non-functional in the DMS assay. Features are ordered according to the proportion of variance in functional scores explained by each feature individually ($R^2$), calculated using linear regression against the experimentally measured functional scores. This highlights the relative explanatory power of the computational features. For this analysis, the FlexddG data used was computed using MTX as a ligand.
(TIFF)

**S7 Fig. Waterfall plots obtained from the SHAP explainer values for mutants of interest.** A) Waterfall plots detailing classification decisions for mutants that were classified as false negatives in the testing dataset for MTX resistance (leading to resistance experimentally but not being predicted as leading to resistance by the model.) V11E (Prediction probability of leading to MTX resistance of 0.33) and V11M (0.36) are both predicted as being destabilizing by MutateX, S37P (0.36) misclassification is mainly driven by its classification as a high confidence allosteric mutation by [44], and G124P (0.25) was classified as being non-functional. B) Waterfall plots detailing classification decisions for mutants that were predicted by the models as leading to TMP resistance. I10K (probability of 0.94), F121H (0.52), I123G (0.82), and Y197Q (0.81) are all novel predictions made by the model (Not measured/predicted as leading to MTX resistance but predicted as leading to TMP resistance.) C) Waterfall plots detailing classification decisions for mutants that were observed either in this or in previous study and have been linked to TMP resistance in PjDHFR. F36C (Prediction probability of leading to TMP resistance of 0.91) was predicted by the model as leading to TMP resistance (and not observed as leading to MTX resistance), but not L65P (0.33), A67V (0.03) or C166Y (0.01). L65P was experimentally identified as being non-functional. A67V was classified as not leading to TMP resistance mainly because of its buriedness level, as well as effect on protein stability. C166Y was classified as not leading to TMP resistance because of its distance to the ligand, buriedness level and effect on protein stability. All SHAP values are calculated on z-score scaled data.
(TIFF)

**S8 Fig. Visualization of model performance when function-related features are taken out of the model.** A) Confusion matrices and ROC/PRD curves for train and testing set splits, with SHAP value explainer plot, with relative impact of features on model decision. True positives and false negatives stayed mostly consistent with the complete model, but much more false positives were predicted, highlighting the importance of functional information. B) Scatterplot comparing ground truth from MTX DMS screening vs model's predicted probability. True negatives in grey, true positives in blue, false positives in yellow and false negatives in orange. Mutations that were identified as functional in the function DMS screening have round markers and mutations identified as non-functional have X-shaped markers. X-shaped markers are much more numerous in the false positive quadrant. C) Heatmap with prediction probability of being resistant to MTX. Mutations with probability >0.5 are identified as resistance conferring, and mutations ≤ 0.5 are classified as sensitive.
(TIFF)

**S9 Fig. Interquartile range of the change in distance between the residue's α-carbon and the closest heavy atom on the ligand.** For each structure generated for all mutations during FlexddG saturation mutagenesis, the median position of the α-carbon of the wild-type structures was used as the baseline. This was compared to the position of the corresponding α-carbon of the mutated residue of all 100 structures for each mutant, resulting in a distribution of changes in distance

between the wild-type α-carbon and the corresponding mutant's α-carbon. From this distribution, we extracted the inter-quartile range of the distribution, representing the "flexibility" of the protein region caused by the mutation. A wide distribution (high IQR) means the mutated residue increases region flexibility, as the mutated region can occupy a broader range of positions further from its wild-type position.
(TIFF)

**S10 Fig. Summary figures of regression-based model performance.** Regression models trained in parallel to their classification counterparts and compared to the best performing model's classification of mutants. A) For a simple Linear Regression model, B) a Support Vector Regression (SVR) model and C) for a Random Forest Regression (RFR) model. The best performing model was the RFR model, as with classification. The RFR model was more consistently linear, but despite predicting less false positives, also tended to yield more false negative predictions. All correlations are Spearman's $\rho$.
(TIFF)

**S11 Fig. Nucleotide substitutions regarding the RU7 reference sequence found in PjDHFRs isolated from clinical cases in Eastern Canada.** A) Position and count of all mutations found during sequencing of 531 PjDHFRs. The y axis is discontinuous because of the high prevalence of a silent mutation at position 90. The track at the top represents secondary structures along the sequence, with α-helices in light grey and beta strands in dark grey. Only the coding sequence (spliced sequence) was considered for this representation. B) Positions and counts of all mutations found in samples where more than one PjDHFR allele was observed. Most mutations identified were accompanied by a wild-type allele. C) Boxplot representing the different mutation types observed as well as their accompanying qPCR counts. No mutation type has significantly greater qPCR counts than the distribution counts from wild-type sequences (Welch's one-sided t-test, unequal variance.).
(TIFF)

## Acknowledgments

We would like to thank members of the Landry lab for their helpful comments throughout the project. We also thank Mélanie Côté for her work in acquiring and sorting the clinical samples.

## Author contributions

**Conceptualization:** Francois D. Rouleau, Alexandre K. Dubé, Christian R Landry.

**Data curation:** Francois D. Rouleau.

**Formal analysis:** Francois D. Rouleau, Alicia Pageau.

**Funding acquisition:** Francois D. Rouleau, Philippe J. Dufresne, Christian R Landry.

**Investigation:** Francois D. Rouleau, Alexandre K. Dubé, Lyne Désautels, Christian R Landry.

**Methodology:** Francois D. Rouleau, Alexandre K. Dubé, Christian R Landry.

**Project administration:** Francois D. Rouleau, Alexandre K. Dubé, Philippe J. Dufresne, Christian R Landry.

**Resources:** Philippe J. Dufresne, Christian R Landry.

**Software:** Francois D. Rouleau, Alicia Pageau.

**Supervision:** Alexandre K. Dubé, Philippe J. Dufresne, Christian R Landry.

**Validation:** Francois D. Rouleau, Alexandre K. Dubé, Christian R Landry.

**Visualization:** Francois D. Rouleau.

**Writing – original draft:** Francois D. Rouleau.

**Writing – review & editing:** Francois D. Rouleau, Alexandre K. Dubé, Alicia Pageau, Lyne Désautels, Philippe J. Dufresne, Christian R Landry.

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
