## [Decision Letter · Decision Letter 0]

19 Jan 2026

PGENETICS-D-25-01212

Predicting antifolate resistance in the unculturable fungal pathogen Pneumocystis jirovecii

PLOS Genetics

Dear Dr. Landry,

Thank you for submitting your manuscript to PLOS Genetics. After careful consideration, we feel that it has merit but does not fully meet PLOS Genetics's publication criteria as it currently stands. Therefore, we invite you to submit a revised version of the manuscript that addresses the points raised during the review process.

We look forward to receiving your revised manuscript.

Kind regards,

Adrian Serohijos, PhD

Guest Editor

PLOS Genetics

Geraldine Butler

Section Editor

PLOS Genetics

Aimée Dudley

Editor-in-Chief

PLOS Genetics

Anne Goriely

Editor-in-Chief

PLOS Genetics

**Additional Editor Comments:**

Dear Dr. Landry

We received comments from two new reviewers, as well as taking into consideration the previous round of reviews at PLoS Pathogens, we have decided to invite you for a major revision.

Specifically, I hope you can address the comment of Reviewer 1 for more information on the modelling and implications of the model validation and generalizability. Reviewer 2 also has minor comments on the interpretation of results.

Please find below their detailed reviews.

**Journal Requirements:**

At this stage, the following Authors/Authors require contributions: Christian R Landry, Alexandre Dubé, Alicia Pageau, Lyne Désautels, and Philippe Dufresne. Please ensure that the full contributions of each author are acknowledged in the "Add/Edit/Remove Authors" section of our submission form.

The list of CRediT author contributions may be found here: https://journals.plos.org/plosgenetics/s/authorship#loc-author-contributions

- ® on page: 30.

5) We notice that your supplementary Figures are included in the manuscript file. Please remove them and upload them with the file type 'Supporting Information'. Please ensure that each Supporting Information file has a legend listed in the manuscript after the references list.

Potential Copyright Issues:

i) Figure 1A. Please confirm whether you drew the images / clip-art within the figure panels by hand. If you did not draw the images, please provide (a) a link to the source of the images or icons and their license / terms of use; or (b) written permission from the copyright holder to publish the images or icons under our CC BY 4.0 license. Alternatively, you may replace the images with open source alternatives. See these open source resources you may use to replace images / clip-art:

7) Thank you for stating 'The sequencing data produced for this work is accessible through SRA BioProjects: PRJNA1243435 and SAMN47937957.'. We found that this accession numbers directs to a 'dataset not found' page. Kindly fix it to point to an active link.We therefore suggest that you provide this information now, though we will not hold up the peer review process if you are unable.

8) Please amend your detailed Financial Disclosure statement. This is published with the article. It must therefore be completed in full sentences and contain the exact wording you wish to be published.

9) Please send a completed 'Competing Interests' statement, including any COIs declared by your co-authors. If you have no competing interests to declare, please state "The authors have declared that no competing interests exist". Otherwise please declare all competing interests beginning with the statement "I have read the journal's policy and the authors of this manuscript have the following competing interests"

**Reviewers' comments:**

Reviewer's Responses to Questions

**Comments to the Authors:**

Reviewer #1: SUMMARY

This study presents a framework for predicting antifolate resistance in Pneumocystis jirovecii, an unculturable fungal pathogen of major clinical consequence. By integrating a large deep mutational scanning data set, biophysical modeling, and machine-learning approaches, the authors train a predictive model for methotrexate resistance and extend it to predict trimethoprim resistance, which cannot be experimentally assayed in vivo. The work is technically strong, conceptually well motivated, and addresses a genuine gap in our ability to study resistance evolution in hard-to-study pathogens.

More importantly, this study presents a responsible use of machine learning methodology in a world where I see too many papers that utilize machine learning in a careless manner. This could serve as a standard for how we use these approaches moving forward. That said, there are some significant challenges with the study in its current form.

My major concerns with this study are that it is unfocused. It reads largely as a technological expedition, which there is nothing wrong with in the pure sense. However, if this is a methods paper, then it should be written as such. And a methods paper often requires that a method be tested in different circumstances, across different data sets.

Alternatively, if this is a paper about genetics or evolution, then it should be written as such. I recognize that the inability to validate the predictions are an impediment to this paper’s biological conclusions. And I don’t want to be the sort of reviewer that makes this sort of demand (which would be a tremendous amount of work), but something must be done to make this paper much more than an interesting machine learning exercise that may have some biological implications. As it stands, the study does a very poor job of teaching me anything in particular about the natural world.

For example, the conclusion reads as if the paper is an engineering proof-of-concept. What, specifically, about the biology of drug resistance, did we learn? Is the point that mutation effects in a different drug can be predicted using a large-scale DMS data set, and machine learning algorithms are used? What does this say about biology and genetics other than that machine learning allows us to predict things? I think this study is successful in using an original approach (the biophysical components, for example, are great). But there is no narrative and the paper isn’t organized in a manner that allows the reader to follow any particular theme.

The authors do discuss some specific mutations and their character. The biophysical features are interesting. But the results and discussion sections overall read as sporadic takes on various findings, mostly focusing on the ability of ML to predict mutations. What does this say for predictive evolution of resistance? Or for drug design? Or for how we think about tradeoffs in protein evolution?

The study need not focus on one thing, but it currently reads as if a large technological feat was pulled off (impressive) and the authors then searched for a relevant question, after the results came in. Again, this is not a bad thing, but as written, the study doesn’t read with the sorts of cohesion that it could.

I suggest the authors write the manuscript from the perspective of the challenges in identifying cross-resistance. And/or the relevance of the findings for medicinal chemistry, or fields interested in identifying how evolution affects resistance mechanisms across drugs. I think this can be achieved with different foci.

Relatedly, I think the study would benefit tremendously from a table that orients the reader to their choice of ML protocol. Too often I see these papers where the method is used and it is not clear to the reader why a given algorithm was used. Here is what I suggest:

A table or box that explains what random forest does, why it is appropriate for the data structure, and substantive scholarly for this. In addition, the authors should highlight other ML methods that have been used for other related studies, and why they do not capture what is necessary for the specific questions posted.

Moreover, the study should have a different summarizing table that highlights the specific biological points, and implications. This would, of course, be tied to rewrites that better organize the study into its methodological and biological implications.

OTHER POINTS:

The abstract is very long and confusing. I’d almost recommend a clinical-style abstract where the authors delineate exactly what the question is, what methods they used, the most notable results, and implications. I actually think this should be standard practice in all machine-learning style papers, as it clearly delineates the data science/computational aspects from the true basic science essence. But this is especially true for this study, where the abstract is difficult to follow.

VERY MINOR COMMENTS:

Line 105: “The DHFR” , Remove “The” (I think)

Abstract: 100-folds should be 100-fold

LINE 175: I could be wrong, but “temperature sensitive (TS) DHFR constructions” should be “temperature sensitive (TS) DHFR constructs.”

Line 237: validating it’s a high degree of importance' should be 'validating its high degree of importance.

Figure 1 caption: PBD: 3CD2 should be PDB: 3CD2

Line 383: selection coefficient a 37 °C should be 'selection coefficient at 37 °C.

Line 595: LOFs mutations is wrong, I think

Reviewer #2: The manuscript by Rouleau et al. uses deep mutational scanning plus structural modeling and machine learning to predict which mutations in Pneumocystis jirovecii dihydrofolate reductase (PjDHFR) can confer resistance to antifolate drugs, especially trimethoprim (TMP), in a pathogen that cannot be cultured in vitro. Because P. jirovecii cannot be grown in the lab, the authors express its DHFR in yeast and perform deep mutational scanning to measure how all single amino‑acid changes affect enzyme function and resistance to methotrexate (MTX), a related antifolate. The work nicely combines experimental DMS for function and MTX resistance, multiple structure‑based predictors (MutateX/FoldX, FlexddG, GEMME), and machine learning into a coherent pipeline tailored to an unculturable pathogen. The analysis identifies distance to ligand, functional status, buriedness, stability, and flexibility as key features, which are aligned with biochemical intuition.

Overall, I found the work to be technically strong, clearly written, which addresses a clinically relevant problem with a workaround to experimental intractability. The integration of functional DMS data as a key feature is a major strength and differentiates this work from prior purely in silico approaches. However, several conceptual and interpretational issues should be addressed to strengthen the manuscript and align claims more tightly with the evidence presented. I will list a few concerns which I hope to help authors strengthen this interesting work.

Major comments/suggestions:

1) While the manuscript emphasizes predictive performance, the most compelling contribution is arguably mechanistic insight into resistance determinants, not prediction alone. Otherwise, one would expect authors to have checked several ML methods including deep neural networks. The random forest model is repeatedly justified by accuracy metrics (balanced accuracy, ROC, PRD), but the biological interpretation of feature importance (distance to ligand, flexibility, functionality) is where the work truly advances the field. I recommend reframing parts of the Results and Discussion to emphasize that the ML model is primarily a tool for integrating heterogeneous and non-linear mechanistic features, rather than a black-box predictor.

2) The central assumption that MTX resistance mechanisms generalize to TMP resistance because both are antifolates is plausible. Although structural similarity and shared binding sites are discussed, the manuscript would benefit from a clearer articulation of where this analogy might break down. In light of the Popperian falsifiability, is there a reason to believe that such transferability might not work?

3) The authors correctly discuss class imbalance. However, the decision threshold of 0.5 for resistance classification is somewhat arbitrary given the severe imbalance (≈3% resistant). Would it make sense to consider stratifying predictions into high-, medium-, and low-confidence classes and explicitly be cautious against over-interpreting marginal predictions?

4) One of the most important findings is that DMS data dramatically improves resistance prediction. This point has implications far beyond P. jirovecii and deserves stronger emphasis. I might have overlooked but the manuscript could more clearly state that purely computational resistance predictors are fundamentally limited without functional data.

Specific comments in relation to the manuscript’s content:

5) The authors state:

“We define these amino acid substitutions as loss-of-function (LOF) mutations.”

This terminology is potentially too strong and may be misleading. In the broader genetics literature, loss-of-function typically refers to classical alleles—such as truncating, nonsense, or frameshift variants—that abolish gene function. In contrast, the variants described here are missense substitutions whose classification as LOF is based on reduced fitness in a specific functional complementation assay in S. cerevisiae under defined temperature and expression conditions. As such, these mutations are more accurately described as strongly deleterious in this assay, or functionally impaired, rather than bona fide LOF alleles.

6) The figure legends are currently insufficient to fully explain the information presented in the figures. Ideally, each figure should be interpretable on its own, solely by reading the legend; however, in several cases this is not possible. For instance, in Figure 2, the track immediately below the heatmap appears to encode contacts between PjDHFR and different ligands, using multiple colors, but this is neither clearly explained nor justified in the legend. Moreover, the same or similar colors are later reused to represent different categories of amino acid substitutions, which may lead to confusion. I recommend that the authors carefully revise all figure legends to ensure they comprehensively describe each visual element and use consistent, unambiguous color schemes, so that the figures are easy to interpret at a glance.

7) The opening of the Results section revisits concepts that are fundamental and already well established in the literature. For example, the observation that highly conserved residues are more likely to yield deleterious substitutions is a long-standing principle in functional variant interpretation, dating back at least to methods such as SIFT. While I am not suggesting that this section be shortened, the authors may reconsider the emphasis and highlight aspects of their results that are more conceptually interesting or novel. One particularly compelling point is the explicit requirement that resistance-conferring mutations must preserve a minimal level of protein function. As the authors note, even mutations that strongly disrupt inhibitor binding will not confer resistance if they simultaneously abolish essential enzymatic activity, as such variants cannot support cell viability. This functional–resistance constraint is less explicitly discussed in much of the existing literature and could be more strongly emphasized as an important conceptual insight emerging from their dataset.

8) The rationale starting around line 298 provides a thorough and well-reasoned rationale for why no single modeling approach can fully capture protein dynamics and function, and why different methods are expected to explain complementary aspects of the DMS results. However, this framing naturally raises the question of the most appropriate statistical framework to quantify these contributions. In the current version, this is supported by correlations, which, while informative, offer a limited view of explanatory power. It may therefore be valuable to complement these analyses with more quantitative approaches, such as estimating the proportion of variance in functional scores explained by each modeling method (e.g., via regression or variance partitioning), or performing enrichment analyses to test whether mutants with high or low functional scores are overrepresented among variants predicted as deleterious or destabilizing by each method. Simple contingency-table–based tests (e.g., Fisher’s exact test) could provide an intuitive and statistically grounded way to assess the overlap between experimental outcomes and model predictions, and would strengthen the link between the conceptual argument and the quantitative evidence.

9) In the section that starts on the random forest model (lines 334 onwards), authors write:

“From various experiments and computational analyses, we believed that we had gathered comprehensive data on different aspects of protein function, resistance, and mechanisms.“

However, the previous section ultimately concludes with the observation that protein stability appears to be the strongest single predictor of DMS-derived functional scores. A more effective way to introduce the subsequent section might be to frame the use of the random forest model as a means of capturing non-linear interactions among multiple features, rather than as a tool primarily aimed at maximizing predictive performance. More generally, while the predictive accuracy of machine-learning models is clearly valuable, their greatest strength in this context may lie in their potential to illuminate underlying biological mechanisms and functional constraints, which I also raised as a major comment. Emphasizing this interpretability-driven perspective, rather than focusing predominantly on predictive power, could strengthen the conceptual framing of this section—though this may ultimately be a matter of emphasis and preference. In this light, it would also be helpful for the authors to explicitly justify their choice of a relatively interpretable machine-learning method such as random forest, as opposed to more opaque but potentially more predictive approaches (e.g., deep neural networks).

10) In several places, the term “resistance” is used interchangeably for biochemical affinity changes and organism-level resistance. Would it make sense to consider clearer distinctions? Also in places, it is written that “mutations are identified as resistant” which I would change to "resistance conferring”.

11) Figure 4 is conceptually very important but is not presented well and gives the impression that it is made in a hurry. It would be nice to show a few residues, compare such residues with the MTX resistance pattern and also map them to the protein structure.

**Have all data underlying the figures and results presented in the manuscript been provided?**

Reviewer #1: None

Reviewer #2: Yes

PLOS authors have the option to publish the peer review history of their article (what does this mean?). If published, this will include your full peer review and any attached files.

Reviewer #1: No

Reviewer #2: No

**Figure resubmission:**
---

## [Decision Letter · Decision Letter 1]

12 May 2026

Dear Dr Landry,

We are pleased to inform you that your manuscript entitled "Predicting antifolate resistance in the unculturable fungal pathogen Pneumocystis jirovecii" has been editorially accepted for publication in PLOS Genetics. Congratulations!

Yours sincerely,

Adrian Serohijos, PhD

Guest Editor

PLOS Genetics

Geraldine Butler

Section Editor

PLOS Genetics

Aimée Dudley

Editor-in-Chief

PLOS Genetics

Anne Goriely

Editor-in-Chief

PLOS Genetics

BlueSky: @plos.bsky.social

Comments from the reviewers (if applicable):

Dear Dr. Landry,

We are recommending your article for publication. We regret the delay as we have been waiting for a reply from Reviewer 1, who eventually opted out of the review.

Looking very closely at the reviews and your replies, including the comments from Reviewer 2, we are of the opinion that all remaining issues have been addressed satisfactorily.

Sincerely,

Adrian Serohijos

Reviewer's Responses to Questions

**Comments to the Authors:**

Reviewer #2: The authors addressed all my raised concerns and the paper is now suitable for publication. I congratulate the team on this nice work.

**Have all data underlying the figures and results presented in the manuscript been provided?**

Reviewer #2: Yes

PLOS authors have the option to publish the peer review history of their article (what does this mean?). If published, this will include your full peer review and any attached files.

Reviewer #2: No

**Data Deposition**

http://datadryad.org/submit?journalID=pgenetics&manu=PGENETICS-D-25-01212R1

**Press Queries**

---

## [Editor Report · Acceptance letter]

PGENETICS-D-25-01212R1

Predicting antifolate resistance in the unculturable fungal pathogen Pneumocystis jirovecii

Dear Dr Landry,

We are pleased to inform you that your manuscript entitled "Predicting antifolate resistance in the unculturable fungal pathogen Pneumocystis jirovecii" has been formally accepted for publication in PLOS Genetics! Your manuscript is now with our production department and you will be notified of the publication date in due course.

With kind regards,

Livia Horvath

PLOS Genetics

On behalf of:
